# Dietary supplementation with compound microecological preparations: effects on the production performance and gut microbiota of lactating female rabbits and their litters

Chengcheng Zhao,[1] Youhao Li,[1] Hui Wang,[1] Ahamba Ifeanyi Solomon,[1] Shuhui Wang,[1] Xianggui Dong,[1] Bing Song,[1] Zhanjun Ren[1]

**ABSTRACT** Early weaning is frequently accompanied by a significant increase in diarrhea and mortality rates, which reduces rabbits' performance. Although antibiotics can reduce pathogenic bacteria, they also harm beneficial microorganisms and disrupt the normal intestinal microbiota balance. In order to find non-residue and non-toxic alternatives to antibiotics to ensure the safety of animal products, we conducted a study on the effect of compound microecological preparations supplementation on lactating female rabbits and their offspring. A total of 60 female rabbits were randomly assigned to four groups: CON, supplemented with probiotics at 3, 6, and 9 g/female rabbit/day from day 24 of gestation until weaning. We observed that probiotics supplementation significantly enhanced production performance ($P < 0.05$), immune and antioxidant function ($P < 0.05$), as well as intestinal flora composition in lactating rabbits and their offspring. Notably, compared with the control group, the experimental group exhibited a 19.23%, 44.22%, and 24.57% increase in milk yield ($P = 0.002$). Regarding rabbit growth performance, the average body weight of young rabbits in the experimental group showed a significant increase of 3.59%, 10.22%, and 6.74% at day 35 ($P = 0.022$), whereas the average daily gain (ADG) of rabbits aged between 21 and 35 days was significantly elevated by 4.94%, 17.06%, and 6.28% in the experimental group ($P < 0.001$). In conclusion, probiotics supplementation can significantly enhance lactation performance, promote growth and disease resistance in rabbits, as well as improve intestinal health when administered at a dosage of 6 g/day. Moreover, the limited sample size in this study may hinder the detection of subtle effects, and augmenting the sample size will bolster the reliability of the study findings.

**IMPORTANCE** The intestinal environment of rabbits is fragile and susceptible to environmental influences, leading to inflammatory intestinal diseases. Adding antibiotics to rabbit feed can achieve the effect of preventing and treating inflammation, which can also lead to the imbalance of the gut microbiota and residual antibiotics in agricultural products. Composite probiotics are live microbial feed additives composed of various ratios of probiotics and have become the most promising alternative to antibiotics due to their residue-free and non-toxic properties. The aim of this study was to investigate the impact of compound probiotics on lactating female rabbits and their offspring. Our findings highlight the potential of compound microecological preparations as an effective strategy for enhancing lactation performance, immune function, and antioxidant capacity in rabbits. The supplementation of probiotics through rabbit milk offers a promising approach to optimize the growth and health outcomes of newborn rabbits.

**KEYWORDS** compound microecological preparation, production performance, gut microbiota, lactating rabbit, intestinal morphology

Address correspondence to Zhanjun Ren, Renzhanjun@nwsuaf.edu.cn, or Bing Song, bingsong@nwafu.edu.cn.

The authors declare no conflict of interest.

With the expansion of large-scale intensive farming, the prevalence of intestinal diseases has emerged as a primary constraint on the growth of the rabbit industry (1). The impairment of intestinal barrier function elevates the likelihood of pathogenic bacteria adhering to the intestinal mucosa, thereby disrupting the dynamic equilibrium among various microorganisms in the digestive tract and between microorganisms and host, increasing susceptibility to intestinal diseases and diarrhea (2), which may ultimately lead to the reduction in the intestine's capacity to absorb and utilize nutrients and a decline in the body's defenses. Meanwhile, early weaning of young rabbits is frequently accompanied by a notable increase in both diarrhea incidence and mortality rate (3, 4), which also poses a significant challenge to the reproductive efficiency of female rabbits (5).

Although the addition of antibiotics to animal feed can partially inhibit pathogenic bacteria, it also disrupts the balance of intestinal microorganisms by killing some beneficial symbiotic microorganisms. In addition, the antibiotic residue problem also poses a safety hazard to livestock products (6–8). As such, it is essential to find non-residue and non-toxic alternatives to antibiotics to ensure the safety of animal products. According to some research findings, yeast, active bacteria, or bacterial spores have demonstrated the potential to prevent enteric diseases in rabbits (9, 10). *Bacillus subtilis* has the ability to secrete surfactants that possess antibacterial activity, thereby enhancing intestinal antioxidant status, fortifying the stability of gut microbiota and regulating its composition in laying hens for maintaining optimal intestinal health (11). Zhao et al. (12) discovered that the regulation of intestinal microbiota diversity and composition by *B. subtilis* can effectively mitigate necrotizing enteritis lesions and growth performance decline induced by *Clostridium perfringens*. *Bacillus licheniformis* is a bacterium of significant commercial value due to its ability to synthesize a diverse range of bacteriocins, antimicrobial peptides, and digestive enzymes, and these bioactive compounds can effectively inhibit pathogenic microorganisms, enhance the stability of intestinal flora, optimize feed nutritional composition, and facilitate efficient digestion and absorption processes within the host organism. The findings of various studies have demonstrated that dietary supplementation with *B. licheniformis* can effectively enhance growth performance, improve heat stress tolerance, and exert preventive effects against necrotizing enterocolitis and coccidiosis in broilers (13, 14). *Saccharomyces cerevisiae* is abundant in easily absorbable protein, nucleotides, amino acids, B vitamins, enzymes, and other essential nutrients (15), which exhibits regulatory effects on livestock immunity and antioxidant function while enhancing production performance, and it has gained extensive utilization in the field of livestock farming (16). In addition, a study has shown that dietary supplementation with *B. subtilis* in rabbits resulted in improved growth performance, intestinal homeostasis, immune organ index enhancement, and increased innate immune response and disease resistance (17). Similarly, oral administration of a recombinant *S. cerevisiae* vaccine was found to elicit an immune response in rabbits, as reported in a previous study (18).

Although some single strains of probiotics can effectively treat diseases of the digestive tract, relevant studies have shown that probiotics from multiple strains may be more effective in broadening the spectrum of protection of microbial infection, and their combined effect has a better inhibitory effect on intestinal pathogens (19). Compound microecological preparations are live microbial feed additives that comprise lactic acid bacteria, *Bacillus*, yeast, and other probiotics in varying proportions, whose inclusion in animal diets can provide similar effects to antibiotics while minimizing environmental risk and safety concerns (20). Probiotics can stimulate the growth of healthy flora and maintain the balance of intestinal microbial communities to prevent the colonization of intestinal pathogens and reduce the occurrence of intestinal diseases (21). Therefore, for farm animals, compound microecological preparations can be considered as a good choice in anti-antibiotic products (22, 23). Furthermore, studies have demonstrated that probiotic supplementation (200 mg of *B. licheniformis* and *B. subtilis*) in conjunction with a fasting regimen can ameliorate the hormonal perturbations induced by postweaning

stress in young rabbits (24). Similarly, a mixture of *B. subtilis* has been found to mitigate injury caused by pathogenic enterotoxigenic *Escherichia coli* (ETEC) and enhance disease resistance in weaning rex rabbits by improving specific members of the intestinal microbiota and immunity (25). Supplementation with a combination of *B. subtilis* and *Lactobacillus acidophilus* has been shown to enhance the populations of beneficial gut bacteria, improve nutrient digestibility, increase cecal fermentation, optimize feed efficiency, and promote growth performance in rabbits (26). In summary, microecological preparations as an alternative to antibiotics have significant effects on improving growth performance, preventing and treating intestinal diseases, and enhancing immunity in rabbits (27, 28).

Although *S. cerevisiae*, *B. subtilis*, and *B. licheniformis* have been extensively investigated in livestock breeding, such as rabbits, there is a paucity of research on the combined application of these three probiotics. We hypothesize that supplementation of compound microecological preparations comprising *B. subtilis*, *S. cerevisiae*, and *B. licheniformis* during lactation could enhance the performance and intestinal microbiota of lactating female rabbits and their offspring, thereby improving the reproductive efficiency of female rabbits and growth outcomes in offspring. Thus, this study aims to compare the effects of incorporating varying levels of compound microecological agents into diets and investigate their impact on the performance, immune response, antioxidant properties, and intestinal flora of both mother rabbits and their offspring, in order to provide theoretical evidence for the application of microecological agents in rabbit breeding production.

## MATERIALS AND METHODS

### Animals and materials

This study was conducted at Enyi Professional Breeding Cooperatives, Baoji, Shaanxi, China. At 24 days of gestation, a cohort of 60 healthy female rabbits with comparable parity and weight were selected for the study (Table 2). The compound microecological preparation (Fushite Biotechnology Co. Ltd, Yangli, China) used in this study was composed of *B. subtilis*, *B. licheniformis*, and *S. cerevisiae*. It was guaranteed to contain at least $2.5 * 10^7$ CFU/g of *B. subtilis*, $2.5 * 10^7$ CFU/g of *B. licheniformis*, and $1.0 * 10^8$ CFU/g of *S. cerevisiae*, respectively. The basic diet was designed according to National Research Council (1977). The nutrition levels of the diets, designed as total mixed ration, were tested. The contents of dry matter (DM), crude protein (CP), and crude fat (EE) were determined according to the procedures of the Association of Official Analytical Chemists (29), whereas neutral detergent fiber (NDF) and acid detergent fiber (ADF) were determined according to Van Soest et al. (30). The N content was determined using an Elementar Vario Macro Cube (Elementar, Hanau, Germany), and the CP content was

TABLE 1  Ingredient and nutrient level of the control diet (%, as fed basis)[a]

| Items | CON | Nutrient levels[c] | CON |
|---|---|---|---|
| Corn | 17.50 | DM | 90.29 |
| Soybean meal | 15.50 | DE (MJ/kg) | 9.47 |
| Wheat bran | 12.00 | CP | 16.62 |
| Wheat middling | 5.00 | EE | 3.01 |
| Grass meal | 44.50 | CF | 17.18 |
| Premix[b] | 5.00 | NDF | 31.57 |
| Soya-bean oil | 0.50 | ADF | 20.32 |
| Total | 100.00 | | |

[a]CP, crude protein; DE, digestible energy; DM, dry matter; EE, crude fat; CF, crude fiber; NDF, neutral detergent fiber; ADF, acid detergent fiber; ash, crude ash.
[b]Provided the following amount of vitamins and minerals per kilogram diet: Fe, 500mg; Cu, 50mg; Zn, 500mg; Mn, 400mg; 120,000IU vitamin A; 40,000IU vitamin D3; 250mg vitamin E.
[c]The digestive energy of nutrients is calculated, the rest are measured values.

**TABLE 2** Effects of compound microecological preparations on lactation performance of female rabbits[a]

| Item | CON | Group A | Group B | Group C | SEM[c] | P value[b] |
|---|---|---|---|---|---|---|
| Parity | 4.5 ± 0.5 | 4.25 ± 0.25 | 3.75 ± 0.48 | 4.00 ± 0.58 | 0.452 | 0.705 |
| Body weight, g | 5,759.32 ± 25.16 | 5,716.04 ± 46.48 | 5,696.29 ± 24.37 | 5,738.42 ± 32.53 | 32.14 | 0.586 |
| Milk yield, g/day | 183.4 ± 10.89[a] | 218.66 ± 11.8[b] | 264.5 ± 38.09[c] | 228.46 ± 11.12[b] | 8.990 | 0.002 |
| ADFI, g/day | 424.0 ± 32.6 | 433.8 ± 37.9 | 443.2 ± 45.6 | 438.3 ± 43.5 | 19.954 | 0.917 |
| Milk composition, % | | | | | | |
| Fat | 11.45 ± 0.96 | 11.55 ± 0.71 | 11.94 ± 0.62 | 11.84 ± 0.73 | 0.152 | 0.666 |
| Protein | 10.96 ± 0.53 | 10.49 ± 0.92 | 10.38 ± 0.82 | 10.59 ± 0.63 | 0.144 | 0.563 |
| Lactose | 1.96 ± 0.14 | 1.83 ± 0.26 | 1.75 ± 0.26 | 1.65 ± 0.21 | 0.047 | 0.090 |
| Lactation-related hormones | | | | | | |
| PRL, mIU/L | 112.17 ± 13.5 | 128.36 ± 21.06 | 140.45 ± 21.26 | 129.41 ± 10.12 | 6.730 | 0.069 |
| GH, ng/mL | 3.9 ± 0.58[a] | 4.38 ± 0.53[ab] | 4.79 ± 0.6[ab] | 4.93 ± 0.69[b] | 0.243 | 0.033 |
| E2, pg/mL | 44.17 ± 5.09[c] | 38.2 ± 5[bc] | 30.36 ± 5.89[a] | 36.08 ± 2.42[ab] | 1.877 | <0.001 |

[a]Control, basal diet without addition of compound microecological preparations; group A, basal diet + 3 g/female rabbit/day compound microecological preparations; group B, basal diet + 6 g/female rabbit/day compound microecological preparations; group C, basal diet + 9 g/female rabbit/day compound microecological preparations.
[b]Different superscripts within a row indicate significant differences ($P < 0.05$).
[c]SEM, standard error of the mean ($n = 4$, number of replicates; larger sample sizes would enhance the reliability of the findings); milk yield, 21 d of farrowing, separating the female rabbit from their litter and weighing the litter before and after lactation; ADFI, average daily feed intake; PRL, prolactin; GH, growth hormone; E2, estradiol.

calculated by multiplying the N with 6.25. The ingredients and nutritional levels of the control diet are listed in Table 1.

The experimental design and scheme of the animal treatments are presented in Figure 1. The study included a total of 60 healthy female rabbits, which were carefully selected based on their similar birth weight and parity (Table 2). These rabbits were then randomly divided into four groups (15 female rabbits per group, 8 rabbits per litter) and fed with basic diet (CON), basic diet plus compound microecological preparations at 3 g/female rabbit/day (group A), basic diet plus compound microecological preparations at 6 g/female rabbit/day (group B), and basic diet plus compound microecological preparations at 9 g/female rabbit/day (group C), respectively. The compound microecological preparation was mixed with the control diet feed at a dose rate of 15 (group A), 30 (group B), and 45 kg/t (group C), respectively, which was recommended by the manufacturer. The trial phase was conducted from day 24 of gestation until weaning, with three feeding periods (0700, 1200, and 1700) to meet the nutritional requirements

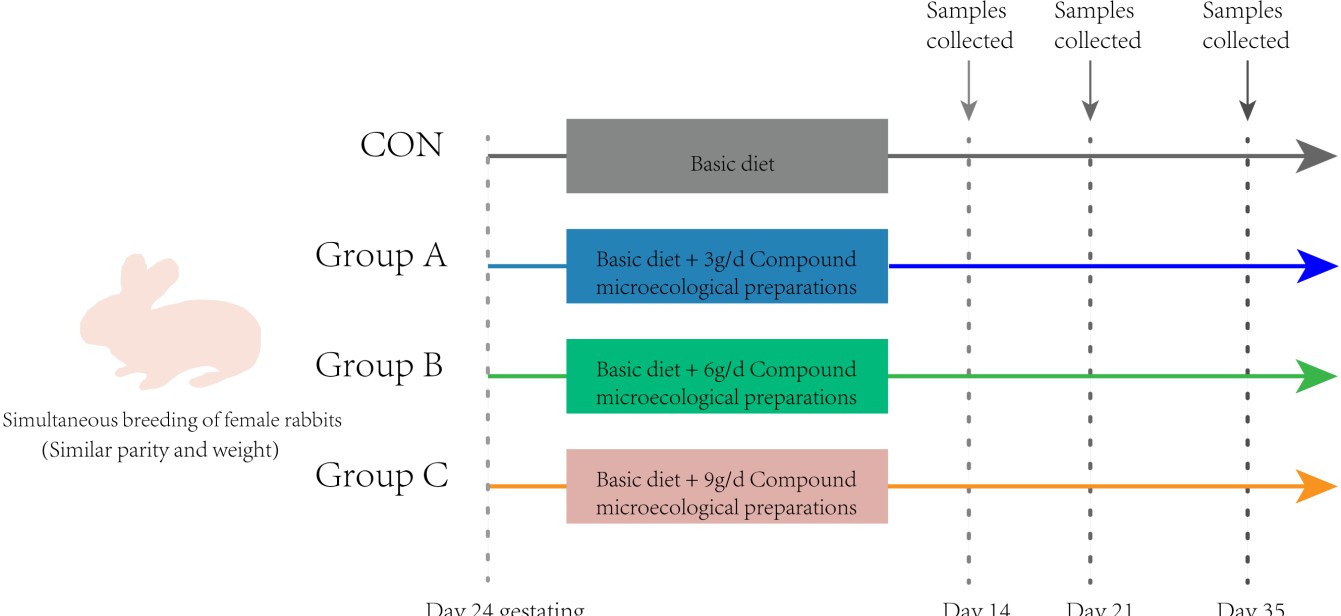

**FIG 1** Experimental design and scheme of the animal treatments.

of female rabbits and ensure continuous access to feed. The daily feed intake of an adult female rabbit is approximately 200 g. However, during pregnancy and lactation, there is a corresponding increase in nutritional demands leading to an elevated consumption rate. In this study, female rabbits in the experimental group were provided with 200 g of specialized diet at 0700 and received additional feedings of basic diet at 1200 and 1700. Throughout the 42-day experimental period, free to feed and water was provided to all female rabbits, with a pre-feeding phase of 7 days followed by a formal feeding phase of 35 days. The average daily feed intake (ADFI) of female rabbits was calculated.

## Sample collection

For young rabbits, after an overnight fasting period at 14 and 35 days of age, four young rabbits were randomly selected from each group for sample collection. Specifically, four litters of rabbits were randomly chosen in each group, and individuals with weights close to the average weight of their respective litter were selected. At the end of the experiment, blood was collected from the heart of the young rabbits, immediately injected into the heparin sodium vacuum blood collection tube after blood collection with a syringe, and mixed upside down for seven to eight times (31). Then, the collected heart blood (5 mL) was centrifuged at 3,000 r/minute for 15 minutes in the laboratory to obtain plasma. This plasma was used for the analysis of biochemical indicators, immune function, and antioxidant status. After the young rabbits were sacrificed by cervical dislocation, the abdomen was cut open immediately, and the jejunum, ileum, and duodenal intestine segments (about 1 cm) were removed respectively, which were gently cleaned with normal saline solution and were fixed in 4% paraformaldehyde for morphological measurement. In addition, after slaughtering the young rabbits, the cecum was carefully isolated and both ends were securely ligated. Subsequently, the cecal contents were aseptically transferred into a 2-mL frozen tube, rapidly frozen in liquid nitrogen, and subsequently stored at −80℃ for microbiological analysis (32).

For female rabbits, on the 21st day of lactation, four female rabbits in each group were randomly selected to collect fresh rabbit breast milk samples for evaluating the content of milk fat, lactose, and milk protein. Before collecting the fresh rabbit breast milk, the back hair around the nipple should be cut with hair scissors as far as possible, and then the breast of the female rabbit should be wiped with a hot towel for 2 minutes. A clean and disinfected cupping machine was used to collect milk under negative pressure, the mouth of the pot was closed to the breast and the nipple was covered to ensure that a closed space is formed in the tank as far as possible, and then repeatedly suctioned to promote milk discharge (33). On the 35th day of lactation, after an overnight fast, 5 mL of blood was collected from the ear vein of the four randomly selected female rabbits in each group. The plasma was then centrifuged at 3,000 r/minute for 15 minutes in the laboratory to obtain samples for subsequent analysis of biochemical indicators, immune and antioxidant markers, and hormonal indicators related to lactation promotion. Most of the healthy rabbits exhibit fecal pellets in their intestinal tract. The specific procedure for collecting feces from female rabbits is as follows: prior to the morning feeding at 0800, one hand firmly grasps the rabbit's ears and neck fur while the palm of the other hand supports its buttocks, causing a slight elevation of the rabbit's abdomen. By gently separating the hind limbs, the anus is exposed. Two fingers are then applied with gentle pressure approximately 5 cm from the anus toward it, resulting in the extrusion of two to three fresh rabbit fecal pellets. The fresh fecal samples were loaded into 5-mL frozen tubes, rapidly frozen in liquid nitrogen, and subsequently stored at −80℃ for further analysis of gut microbiota.

## Evaluation of lactation performance of female rabbits

In terms of the lactation performance of female rabbits, the parameters that require evaluation encompass milk quality (milk fat, lactose, milk protein), milk yield, and hormone expression linked to lactation (34, 35). The levels of lactose, milk fat, and milk protein were measured by MilkoScanFT1 supplied by Yangling Hengman Biotechnology

Development Center (Yangli, China). The milk yield of the female rabbits was quantified on day 21 post-farrowing by isolating the doe from her litter, weighing the litter before and after suckling, and recording the resultant difference in weight as an indicator of lactation performance (36). The levels of hormone expression associated with lactation (lactogen [prolactin, PRL]), growth hormone (GH), and estradiol (E2) were measured by standard commercial kits following the manufacturer's instructions (Quanzhou Ruixin Biotechnology Co., Ltd, Quanzhou, China).

## Evaluation of biochemical, immune, and antioxidant indicators of rabbits

The levels of total protein (TP), albumin (ALB), urea nitrogen (UN), total cholesterol (TCHO), triglycerides (TG), glutamic oxaloacetic transaminase (AST), lactate dehydrogenase (LDH), and glutamate transaminase (ALT) were measured by standard commercial kits following the manufacturer's instructions (Quanzhou Ruixin Biotechnology Co., Ltd, Quanzhou, China).

The levels of immune indicators (immunoglobulin G [IgG], immunoglobulin A [IgA], immunoglobulin M [IgM], secretory immunoglobulin A [sIgA], interleukin-2 [IL-2], interleukin-6 [IL-6], interleukin-8 [IL-8], and tumor necrosis factor-α [TNF-α]) and antioxidant indicators (catalase [CAT], glutathione peroxidase [GSH-PX], superoxide dismutase [SOD], and malondialdehyde [MDA]) were measured by standard commercial kits following the manufacturer's instructions (Quanzhou Ruixin Biotechnology Co., Ltd, Quanzhou, China) (37, 38).

## Evaluation of intestinal morphology and growth performance of young rabbits

Young rabbit jejunum, ileum, and duodenum sections were collected and processed using the traditional paraffin tissue sectioning method, as published previously by Ren et al. (39). Stained tissue sections were then imaged under a BA210 digital microscope from 10 different views per section. Measurements of intestinal villi height and crypt depth were taken, with subsequent calculation of the villi-height-to-crypt-depth ratio (V/C).

During the experiment, the health status of the young rabbits was recorded for each group. The weight of each litter (eight rabbits in a litter) was measured at 0800 at 0, 14, 21, and 35 days of age. The average body weight and average daily gain (ADG) of newborn rabbits were calculated.

## High-throughput 16S ribosomal RNA gene sequencing

Microbiome analysis using 16S rRNA high-throughput sequencing technology after collecting a sample of cecal contents from the rabbits was performed by Beijing Biomarker Technologies Co., Ltd. (Beijing, China). After genomic DNA was extracted from the sample, the quality and quantity of the extracted DNA were examined using electrophoresis on a 1.8% agarose gel, and DNA concentration and purity were determined with NanoDrop 2000 UV-Vis spectrophotometer (Thermo Scientific, Wilmington, USA). The hypervariable region V3-V4 of the bacterial 16S rRNA gene was amplified with primer pairs 338F: 5′-ACTCCTACGGGAGGCAGCA-3′ and 806R: 5′-GGAC TACHVGGGTWTCTAAT-3′. Both the forward and reverse 16S primers were tailed with sample-specific Illumina index sequences to allow for deep sequencing. The PCR was performed in a total reaction volume of 10 µL: DNA template 5–50 ng, forward primer (10 µM) 0.3 µL, reverse primer (10 µM) 0.3 µL, KOD FX Neo Buffer 5 µL, dNTP (2 mM each) 2 µL, KOD FX Neo 0.2 µL, and finally ddH2O up to 20 µL. After an initial denaturation at 95°C for 5 minutes, it was followed by 20 cycles of denaturation at 95°C for 30 seconds, annealing at 50°C for 30 seconds, extension at 72°C for 40 seconds, and a final step at 72°C for 7 minutes. The amplified products were purified with Omega DNA purification kit (Omega Inc., Norcross, GA, USA) and were quantified using Qsep-400 (BiOptic, Inc., New Taipei City, Taiwan, ROC). The amplicon library was paired-end sequenced (2 × 250) on an Illumina novaseq6000 (Beijing Biomarker Technologies Co., Ltd., Beijing, China) (40).

The qualified sequences with more than a 97% similarity threshold were assigned to operational taxonomic units (OTUs) using USEARCH (version 10.0). Taxonomy annotation of the OTUs was performed based on the Naive Bayes classifier in QIIME2 (41) using the SILVA database (42) (release 138.1) with a confidence threshold of 70%. Alpha diversity analysis was performed to identify the species diversity of each sample utilizing QIIME2 software. Beta diversity calculations were performed using principal coordinate analysis (PCoA) to assess the diversity in samples for species complexity. One-way analysis of variance (ANOVA) was used to compare bacterial abundance and diversity. Linear discriminant analysis (LDA) coupled with effect size (LEfSe) was applied to evaluate the differentially abundant taxa. The online platform BMKCloud (https://www.biocloud.net) was used to analyze the sequencing data.

Our study obtained a total of 2,125,018 sequences across all samples in the high-throughput 16S rRNA gene sequencing analysis. The number of sequences per sample ranged from 28,048 to 55,814, with an average of approximately 44,271 sequences per sample. The dada2 method implemented in the QIIME2 2020.6 software was used to perform sequence quality filtering and denoising (41, 43).

## Statistical analysis

The data analysis of this study employed a diverse range of statistical methods to ensure comprehensiveness and accuracy, thereby enhancing the professional and academic quality of the research. The experimental data were initially processed in Microsoft Excel 2019 software, and then the statistical significance was experimentally tested using one-way ANOVA in IBM SPSS Statistics 27 software (SAS Inc., Chicago, IL, USA). Differences between groups were compared using the least significant difference (LSD) method, with $P < 0.05$ considered significant and $P < 0.01$ highly significant (44). The results are presented as mean ± standard error (SEM), and multiple comparisons were corrected using Duncan's correction (45).

It is worth noting that the sample size in this study was limited, which may have implications for detecting subtle effects. Furthermore, the experimental design focused on a specific probiotic formulation and dose, potentially limiting its generalizability to all production environments or rabbit populations. Future studies should consider larger sample sizes and more comprehensive experimental designs to validate these preliminary findings and explore the potential impact of compound probiotics on other animal populations.

## RESULTS

### Effect on lactation performance of female rabbits

Data on the lactation performance of the female rabbits are listed in Table 2. In this study, we conducted a comprehensive analysis on the feed intake of female rabbits to assess their body condition. Our findings revealed no statistically significant differences in average feed intake across all experimental groups ($P > 0.05$). Compared with the control group, there was no significant change in the milk lactose content ($P = 0.090$), milk protein content ($P = 0.563$), and milk fat content ($P = 0.666$) of the rabbit after the addition of the compound microecological preparations. However, it is noteworthy that compared with the control group, the lactation yield of rabbits in groups A, B, and C was increased by 19.23%, 44.22%, and 24.57% ($P = 0.002$), respectively, and the lactation yield of group B was higher than that of the other three groups. In terms of lactation-related hormones, we observed a significant decrease in E2 levels by 13.52%, 31.27%, and 18.32% ($P < 0.001$) in the experimental group compared with the control group, and the growth hormone levels showed a significant increase by 12.31%, 22.82%, and 26.41% ($P < 0.05$), respectively. Besides, no significant differences were observed in prolactin levels among the groups ($P > 0.05$), whereas group B exhibited a significant increase of 25.21% compared with the control group ($P < 0.05$).

## Effect on plasma biochemical indexes of rabbits

The results presented in Table 3 indicate that there were no significant differences observed in the biochemical indicators between the experimental and control groups of female rabbits ($P > 0.05$). The plasma levels of AST, ALT, LDH, TP, ALB, TG, UN, and TCHO in young rabbits did not exhibit any statistically significant differences ($P > 0.05$).

## Effect on the immune and antioxidant indexes of rabbits

The impact of incorporating compound microecological preparations into the diet of the female rabbits on immune indexes is illustrated in Table 4. Compared with the control group, the plasma concentrations of IgG in groups A, B, and C exhibited significant increases of 27.48%, 41.09%, and 29.04%, respectively; the concentrations of IgA showed significant increases of 13.62%, 21.74%, and 22.94%, respectively; and the concentrations of IgM demonstrated significant increases of 21.46%, 21.33%, and 15.21%, respectively ($P < 0.05$). Compared with the control group, the concentrations of sIgA in the plasma of groups A, B, and C exhibited significant increases of 20.84%, 46.89%, and 26.67% ($P = 0.028$), respectively. In terms of cytokines, the addition of compound microecological preparations to the rabbit diet significantly increased the concentrations of anti-inflammatory cytokines IL-2 and IL-8 ($P < 0.05$) while decreasing the levels of pro-inflammatory cytokines IL-6 and TNF-α. Among all groups, group B exhibited the most pronounced effect in elevating anti-inflammatory cytokine concentrations and reducing pro-inflammatory cytokine contents in plasma. Compared with the control group, group B showed a significant increase of 43.67% and 40.28% in the plasma concentrations of anti-inflammatory cytokines IL-2 and IL-8, respectively, along with a significant decrease of 8.28% and 12.23% in contents of pro-inflammatory cytokines IL-6 and TNF-α. In terms of antioxidant indexes, the supplementation of compound microecological preparation did not have a significant impact on the content of SOD in plasma ($P > 0.05$), whereas it significantly affected the concentrations of other antioxidant indexes, including GSH-PX, CAT, and MDA ($P < 0.05$). The concentrations of GSH-PX and CAT in the plasma of female rabbits in group B were significantly increased by 36.97% and 28.12%, respectively, and the concentration of MDA was significantly decreased by 36.86%. In terms of improving the immune performance and antioxidant performance of female rabbits, group B

**TABLE 3** Effects of compound microecological preparations on plasma biochemical markers of rabbits[a,b]

| Item | CON | Group A | Group B | Group C | SEM | P value |
|---|---|---|---|---|---|---|
| Female rabbit | | | | | | |
| AST, nmol/min/mL | 8.14 ± 1.71 | 7.18 ± 0.68 | 8.33 ± 0.73 | 8.19 ± 1.31 | 0.554 | 0.525 |
| ALT, nmol/min/mL | 6.48 ± 1.73 | 5.84 ± 1.71 | 6.13 ± 0.97 | 5.8 ± 2.26 | 0.833 | 0.941 |
| LDH, nmol/min/mL | 11.74 ± 1.81 | 8.7 ± 2.26 | 13.49 ± 3.85 | 10.92 ± 4.48 | 1.549 | 0.273 |
| TP (10%), mg/mL | 24.06 ± 3.17 | 24.97 ± 3.96 | 21.13 ± 1.34 | 22.47 ± 3.31 | 1.472 | 0.351 |
| ALB, g/L | 18.3 ± 6.01 | 18.24 ± 8.03 | 19.12 ± 3.9 | 18.45 ± 2.46 | 2.55 | 0.995 |
| TG, mmol/L | 0.73 ± 0.02 | 0.64 ± 0.11 | 0.67 ± 0.09 | 0.73 ± 0.03 | 0.031 | 0.207 |
| BUN, mmol/L | 4.51 ± 1.59 | 4.51 ± 0.94 | 5.77 ± 0.83 | 4.94 ± 0.81 | 0.522 | 0.356 |
| TCHO, mmol/L | 3.12 ± 0.67 | 3.04 ± 1.04 | 2.49 ± 0.75 | 3.41 ± 0.68 | 0.392 | 0.453 |
| Young rabbit | | | | | | |
| AST, nmol/min/mL | 6.93 ± 1.57 | 5.8 ± 2.45 | 6.93 ± 1.45 | 6.93 ± 0.98 | 0.806 | 0.727 |
| ALT, nmol/min/mL | 5.87 ± 1.13 | 4.69 ± 0.97 | 6.01 ± 1.01 | 6.42 ± 1.13 | 0.53 | 0.175 |
| LDH, nmol/min/mL | 13.87 ± 1.22 | 10.99 ± 0.84 | 13.04 ± 2.51 | 11.19 ± 1.87 | 0.806 | 0.097 |
| TP (10%), mg/mL | 23.01 ± 2.04 | 23.77 ± 2.18 | 22.05 ± 1.38 | 23.43 ± 3.42 | 1.128 | 0.759 |
| ALB, g/L | 16.81 ± 2.87 | 13.22 ± 2.91 | 13.46 ± 2.57 | 14.45 ± 1.94 | 1.286 | 0.244 |
| TG, mmol/L | 0.85 ± 0.11 | 0.61 ± 0.11 | 0.7 ± 0.18 | 0.67 ± 0.16 | 0.07 | 0.182 |
| BUN, mmol/L | 4.96 ± 0.65 | 4.18 ± 0.45 | 3.75 ± 0.59 | 4.35 ± 1.1 | 0.349 | 0.19 |
| TCHO, mmol/L | 3.29 ± 0.47 | 2.88 ± 0.27 | 2.77 ± 0.75 | 2.91 ± 0.44 | 0.243 | 0.535 |

[a]SEM, standard error of the mean ($n = 4$, number of replicates; larger sample sizes would enhance the reliability of the findings); AST, aspartate aminotransferase; ALT, alanine transaminase; LDH, lactate dehydrogenase; TP, total protein; ALB, albumin; TG, triglycerides; UN, urea nitrogen; TCHO, total cholesterol.
[b]Control, basal diet without addition of compound microecological preparations; group A, basal diet + 3 g/female rabbit/day compound microecological preparations; group B, basal diet + 6 g/female rabbit/day compound microecological preparations; group C, basal diet + 9 g/female rabbit/day compound microecological preparations.

**TABLE 4** Effects of compound microecological preparations on the immune and antioxidant indexes of rabbits[c]

| Item | CON | Group A | Group B | Group C | SEM[a] | P value[b] |
|---|---|---|---|---|---|---|
| **Female rabbit** | | | | | | |
| IgG, mg/mL | 12.12 ± 0.86[a] | 15.45 ± 1.34[b] | 17.1 ± 2.02[b] | 15.64 ± 2.17[b] | 0.799 | 0.008 |
| IgA, µg/mL | 193.79 ± 13.98[a] | 220.19 ± 7.37[ab] | 235.92 ± 9.8[b] | 238.25 ± 31.37[b] | 7.815 | 0.017 |
| IgM, µg/mL | 549.99 ± 49.19[a] | 668.02 ± 65.22[b] | 667.31 ± 62.65[b] | 633.62 ± 55.42[ab] | 29.06 | 0.046 |
| sIgA, µg/mL | 22.65 ± 3.53[a] | 27.37 ± 3.61[ab] | 33.27 ± 6.44[b] | 28.69 ± 2.01[ab] | 1.948 | 0.028 |
| IL-2, pg/mL | 58.23 ± 3.16[a] | 77.35 ± 9.59[b] | 83.66 ± 5.07[b] | 80.26 ± 5.83[b] | 2.956 | <0.001 |
| IL-6, pg/mL | 80.33 ± 3.45[b] | 80.57 ± 3.89[b] | 73.68 ± 3.06[a] | 77.73 ± 2.1[ab] | 1.561 | 0.034 |
| IL-8, pg/mL | 62.46 ± 4.95[a] | 72.04 ± 12.48[ab] | 87.62 ± 7.53[b] | 81.51 ± 14.36[b] | 4.915 | 0.027 |
| TNF-α, pg/mL | 163.75 ± 11.23[b] | 141.19 ± 10.83[a] | 143.72 ± 8.05[a] | 142.37 ± 10.06[a] | 5.021 | 0.025 |
| GSH-PX, ng/mL | 88.64 ± 9.81[a] | 98.97 ± 21.08[ab] | 121.41 ± 8.47[b] | 103.88 ± 9.29[ab] | 6.082 | 0.028 |
| SOD, ng/mL | 532.87 ± 38.64 | 546.27 ± 52.95 | 598.15 ± 42.9 | 593.97 ± 67.27 | 25.22 | 0.231 |
| CAT, ng/mL | 43.99 ± 6.1[a] | 46.36 ± 5.62[a] | 56.36 ± 3.57[b] | 50.63 ± 3.56[ab] | 2.357 | 0.018 |
| MDA, nmol/mL | 7.46 ± 0.72[c] | 6.34 ± 0.75[b] | 4.71 ± 0.41[a] | 5.59 ± 0.29[ab] | 0.272 | <0.001 |
| **Young rabbit** | | | | | | |
| IgG, mg/mL | 16.84 ± 2.12 | 19.82 ± 1.2 | 16.28 ± 0.37 | 17.63 ± 3.21 | 0.862 | 0.123 |
| IgA, µg/mL | 242.28 ± 17.32[a] | 259.64 ± 1.51[b] | 232.32 ± 10.97[a] | 245.4 ± 5.38[ab] | 4.398 | 0.024 |
| IgM, µg/mL | 660.26 ± 78.83 | 735.7 ± 48.57 | 635.31 ± 84.74 | 658.49 ± 16.06 | 28.525 | 0.182 |
| sIgA, µg/mL | 33.6 ± 2.57[ab] | 36.95 ± 1.29[b] | 30.5 ± 4.32[a] | 31.75 ± 0.77[a] | 1.12 | 0.023 |
| IL-2, pg/mL | 98.9 ± 6.92 | 104.4 ± 6.09 | 100.95 ± 14.47 | 96.1 ± 15.87 | 5.418 | 0.784 |
| IL-6, pg/mL | 75.82 ± 4.62[c] | 58.81 ± 3.63[a] | 70.58 ± 3.45[bc] | 67.8 ± 7.12[b] | 2.353 | 0.003 |
| IL-8, pg/mL | 63.07 ± 12.52[a] | 60.43 ± 6.64[a] | 92.87 ± 4.87[b] | 71.37 ± 4.3[a] | 3.542 | <0.001 |
| TNF-α, pg/mL | 150.76 ± 17.11 | 120.08 ± 13.02 | 145.78 ± 18.48 | 130.5 ± 12.89 | 7.687 | 0.059 |
| GSH-PX, ng/mL | 112.58 ± 6.63 | 133.98 ± 14.42 | 113.47 ± 8.68 | 116.56 ± 12.01 | 5.217 | 0.053 |
| SOD, ng/mL | 405.01 ± 49.86 | 485.11 ± 60.06 | 463.46 ± 44.43 | 440.67 ± 13.93 | 21.036 | 0.133 |
| CAT, ng/mL | 56.41 ± 6.73[a] | 72.29 ± 2.62[c] | 63.91 ± 1.93[b] | 61.07 ± 3.21[ab] | 1.811 | 0.001 |
| MDA, nmol/mL | 4.45 ± 0.17 | 3.98 ± 0.34 | 3.82 ± 1.34 | 4.21 ± 1.22 | 0.383 | 0.788 |

[a]SEM, standard error of the mean ($n = 4$, number of replicates; larger sample sizes would enhance the reliability of the findings); IgG, immunoglobulin G; IgA, immunoglobulin A; IgM, immunoglobulin M; sIgA, secretion immunoglobulin; IL-2, interleukin-2; IL-6, interleukin-6; IL-8, interleukin-8; TNF-α, tumor necrosis factor-α; GSH-PX, glutathione peroxidase; SOD, superoxide dismutase; CAT, catalase; MDA, malondialdehyde.
[b]Different superscripts within a row indicate significant differences ($P < 0.05$).
[c]Control, basal diet without addition of compound microecological preparations; group A, basal diet + 3 g/female rabbit/day compound microecological preparations; group B, basal diet + 6 g/female rabbit/day compound microecological preparations; group C, basal diet + 9 g/female rabbit/day compound microecological preparations.

achieved the best effect overall, and groups A and B showed advantages only in some indexes of IgA and IgM.

For young rabbits, compared with the control group, the experimental groups A and C exhibited a significant increase in plasma IgA concentration by 7.17% and 1.29%, respectively ($P = 0.024$), and the experimental group A showed a significant elevation of 9.97% in plasma sIgA concentration ($P = 0.023$). In terms of cytokines, the indirect administration of compound microecological preparations through rabbit milk as a medium significantly increased the plasma concentration of anti-inflammatory cytokine IL-8 in young rabbits ($P < 0.001$), and this increase was observed to be 47.25% and 13.16% in groups B and C, respectively. Furthermore, it led to a significant reduction in the plasma content of pro-inflammatory cytokine IL-6 by 22.43%, 6.91%, and 10.58% in groups A, B, and C, respectively ($P = 0.003$). Additionally, there was an inclination toward reducing the concentration of pro-inflammatory cytokine TNF-α in plasma ($P = 0.059$). In terms of antioxidant indexes, the intergroup effects of compound microecological preparations indirectly administered to rabbits through breast milk did not show significant changes in the levels of SOD, MDA, and GSH-PX in plasma ($P > 0.05$). However, there were statistically significant effects on the concentration of CAT in plasma; compared with the control group, groups A, B, and C exhibited a significant increase in CAT concentration by 28.15%, 13.30%, and 8.26%, respectively ($P = 0.001$). The findings suggest that the indirect administration of compound microecological preparations through rabbit milk exerts a significant impact on certain immune and

antioxidant markers in young rabbits while also effectively enhancing their immune and antioxidant capacity to a certain extent (Table 4).

## Effect on the intestinal microbiota of female rabbits

### OTU statistics and diversity analysis

As shown in Table 5, there was no significant difference in OTU number between the experimental and control groups ($P > 0.05$), and the addition of compound microecological preparations to the diets had no significant effect on the Ace, diversity index (Chao1), and Shannon index (Shannon) of bacterial 16S rRNA in the intestinal tracts of the female rabbits ($P > 0.05$). The higher the coverage of the sample library, the higher the probability of detecting the species in the sample (46). Therefore, the coverage value can directly reflect whether the sequencing result represents the actual existence of microorganisms in the sample (47). The sequencing depth in the fresh fecal samples of the female rabbits in all groups was sufficient, as indicated by a sequencing coverage value above 99% for the detected OTUs (Table 5).

According to the Shannon curves of microbial diversity indices constructed from the fresh fecal samples of the female rabbits with different sequencing quantities, it can be seen in Figure 2 that the sequencing curves of all samples were relatively flat, indicating that the sequencing quantities of the samples were sufficient to accurately respond to most of the microbial information of the cecum contents of the rabbits, which basically met the requirements of the experiment. It can basically meet the requirements of the experiment.

### OTU Venn diagram analysis

Figure 2B illustrates the Venn diagram, which displays the number of OTUs that are unique to each of the four sample groups and are shared among them. The degree of overlap in OTUs is visually apparent. In total, there are 5,129 OTUs across all groups, with 260 overlapping among them. The control group was the largest with 1,870 OTUs, of which 1,009 were unique OTUs. Group B had the fewest, with 1,654 OTUs, of which 983 were unique.

### NMDS and PERMANOVA

Nonmetric multidimensional scaling (NMDS) is a kind of ranking method suitable for ecological research, which mainly simplifies the research objects in multidimensional space to low-dimensional space for positioning, analysis, and classification while preserving the original relationship between the objects (48). The points in the figure represent the individual samples respectively, different colors represent different groups, and the distance between points indicates the degree of difference. If stress is less than 0.2, it means that NMDS analysis has some reliability. The smaller the distance between samples on the coordinate plot, the higher the similarity. Permutational multivariate analysis of variance (PERMANOVA) analysis is also called Adonis analysis. It uses the distance matrix to decompose the total variance, analyzes the interpretation of different grouping factors on the sample differences, and uses the permutation test to analyze the statistical significance of the division. NMDS and PERMANOVA based on Bray-Curtis

**TABLE 5** Effect of compound microecological preparations on gut microbial diversity of female rabbits

| Item | CON | Group A | Group B | Group C | SEM[a] | P value |
|---|---|---|---|---|---|---|
| OTU_num | 580.75 ± 91 | 564 ± 59.88 | 525.25 ± 107.34 | 572 ± 55.84 | 39.257 | 0.782 |
| Coverage | 99.97 ± 0.01 | 99.97 ± 0.01 | 99.98 ± 0.01 | 99.98 ± 0.02 | 0.006 | 0.772 |
| Ace | 587.18 ± 89.77 | 570.54 ± 61.47 | 529.37 ± 109.49 | 576.19 ± 59.1 | 39.978 | 0.773 |
| Chao1 | 583.68 ± 89.77 | 567.7 ± 61.44 | 526.94 ± 108.48 | 573.84 ± 58.16 | 39.731 | 0.777 |
| Shannon | 7.59 ± 0.29 | 7.54 ± 0.38 | 7.45 ± 0.44 | 7.53 ± 0.36 | 0.184 | 0.954 |

[a]SEM, standard error of the mean ($n = 4$, number of replicates; larger sample sizes would enhance the reliability of the findings).

distance show that the value of the stress function for gut flora community structure in the female rabbits is 0.19, and the distribution among the four groups is relatively decentralized (Fig. 2C), with no significant differences ($P > 0.05$) (Fig. 2D).

### Analysis of gut microbial community structure at the phylum level

As can be seen from Figure 2E and Table 6, at the level of phylum, the cecal flora of each test group mainly consisted of *Firmicutes*, *Bacteroidetes*, and a *Verrucomicrobia*. There were no significant differences in the abundances of *Firmicutes*, *Bacteroidetes*, and *Verrucomicrobia* among all groups ($P > 0.05$).

### Analysis of gut microbial community structure at the family level

As depicted in Figure 2F and Table 6, at the family level, the difference analysis of the dominant flora revealed no significant differences between the control group and the experimental groups at the *Lachnospiraceae*, *Ruminococcaceae*, and *Oscillospiraceae* levels ($P > 0.05$). However, a significant difference was observed at the *Rikenellaceae* level ($P = 0.024$), where the experimental groups exhibited a significantly higher relative abundance compared with the control group.

### Analysis of gut microbial community structure at the genus level

It can be seen from Figure 2G and Table 6 that at the genus level, there was no significant difference among groups after the difference analysis of the dominant flora ($P > 0.05$). The experimental results also indicated an increasing trend in the relative abundance of *Rikenellaceae_RC9_gut_group* with the changes observed in the experimental groups, although the difference did not reach statistical significance ($P = 0.068$).

## Effect on growth performance of rabbit litters

As shown in Table 7, during the initial period from 0 to 14 days, the average body weight of young rabbits in all groups showed no significant differences ($P > 0.05$). Nonetheless, on day 21, the average body weight of young rabbits in group B demonstrated a statistically significant increase compared with groups CON, A, and C ($P = 0.02$). The

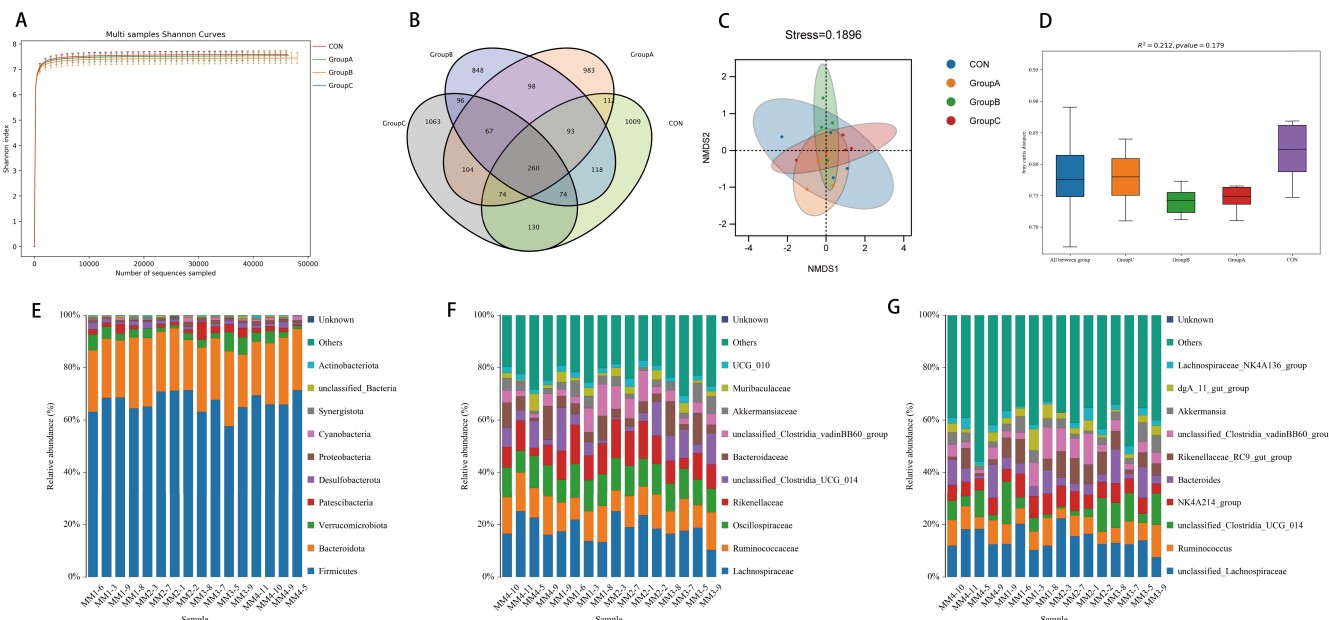

**FIG 2** Effect on the intestinal microbiota of female rabbits. (A) Shannon curve and (B) OTU Venn diagram of the cecal microflora of female rabbits. (C) NMDS and (D) PERMANOVA of the gut microflora of female rabbits. Relative abundance of OTUs about the gut microflora of female rabbits at the (E) phylum, (F) family, and (G) genus levels.

**TABLE 6** Relative abundance of OTUs about the gut microflora of female rabbits

| Item | CON | Group A | Group B | Group C | SEM[a] | P value[b] |
|---|---|---|---|---|---|---|
| Phylum | | | | | | |
| Firmicutes | 68.15 ± 2.72 | 66.13 ± 2.79 | 69.62 ± 3.01 | 63.32 ± 4.22 | 1.594 | 0.083 |
| Bacteroidota | 23.23 ± 2.13 | 23.74 ± 2.32 | 23.02 ± 2.9 | 24.21 ± 3.47 | 1.353 | 0.928 |
| Verrucomicrobiota | 2.83 ± 1.51 | 4.14 ± 1.48 | 2.23 ± 1.12 | 4.73 ± 2.63 | 0.843 | 0.225 |
| Family | | | | | | |
| Lachnospiraceae | 20.05 ± 4.48 | 16.54 ± 3.98 | 21.53 ± 3.33 | 15.78 ± 3.77 | 1.944 | 0.17 |
| Ruminococcaceae | 13.77 ± 1.6 | 11.19 ± 2.2 | 11.05 ± 2.14 | 10.98 ± 2.83 | 1.096 | 0.276 |
| Oscillospiraceae | 11 ± 1.83 | 11.5 ± 1.86 | 11.57 ± 0.74 | 10.33 ± 1.06 | 0.688 | 0.619 |
| Rikenellaceae | 7.53 ± 3.51[a] | 11.76 ± 2.31[ab] | 13.27 ± 1.75[b] | 8.01 ± 2.68[a] | 1.281 | 0.024 |
| Genus | | | | | | |
| UL | 15.22 ± 3.51 | 13.73 ± 4.46 | 16.69 ± 4.07 | 11.71 ± 2.83 | 1.859 | 0.327 |
| Ruminococcus | 8.08 ± 2.44 | 7.86 ± 2.03 | 5.75 ± 1.83 | 8.42 ± 2.9 | 1.148 | 0.397 |
| UCU_014 | 5.9 ± 3.74 | 6.75 ± 6.65 | 4.65 ± 5.6 | 8.96 ± 3.8 | 2.473 | 0.684 |
| NK4A214_group | 5.67 ± 0.95 | 7.5 ± 1.94 | 6.76 ± 1.24 | 5.78 ± 1.49 | 0.702 | 0.282 |
| Bacteroides | 7.2 ± 5.13 | 5.35 ± 2.59 | 3.38 ± 0.63 | 7.51 ± 5.75 | 1.763 | 0.480 |
| RRG_group | 0.03 ± 0.03 | 0.05 ± 0.04 | 0.09 ± 0.01 | 0.05 ± 0.02 | 0.013 | 0.068 |

[a]SEM, standard error of the mean ($n = 4$, number of replicates; larger sample sizes would enhance the reliability of the findings); UL, unclassified_Lachnospiraceae; UCU_014, unclassified_Clostridia_UCG_014; RRG_group, Rikenellaceae_RC9_gut_group.
[b]Different superscripts within a row indicate significant differences ($P < 0.05$).

impact of compound probiotics on the average body weight of young rabbits became more pronounced on the 35th day. In comparison with the control group, experimental groups A, B, and C exhibited a significant increase in average body weight by 3.59%, 10.22%, and 6.74%, respectively ($P = 0.022$). When evaluating the ADG during the initial 0 to 14 days, groups B and C exhibited highly significantly superior gains compared with groups A and CON ($P < 0.001$). Similarly, from 21 to 35 days, the ranking of ADG across all groups was as follows: group B > group C > group A > CON, and the average daily gain of the experimental groups was significantly increased by 4.94%, 17.06%, and 6.28% ($P < 0.001$), respectively. These compelling findings provide evidence that the addition of compound microecological preparations to the diet significantly enhanced the average body weight and ADG of young rabbits, and group B had the best results overall.

## Effect on the intestinal morphology of rabbit litters

According to Figure 3 and Table 8, there were no significant differences observed in the villus height, crypt depth, and intestinal wall thickness of the ileum, jejunum, and duodenum among all groups ($P > 0.05$). Additionally, the V/C value of the ileum in the experimental groups showed an increasing trend in these two stages.

**TABLE 7** Effect of compound microecological preparations on growth performance of young rabbits

| Item | CON | Group A | Group B | Group C | SEM[a] | P value[b] |
|---|---|---|---|---|---|---|
| BW, g | | | | | | |
| 0 day | 78.98 ± 5.17 | 83.88 ± 5.46 | 79.95 ± 2.96 | 84.62 ± 6.45 | 2.05 | 0.186 |
| 7 days | 160.06 ± 18.28 | 155.3 ± 11.22 | 170.13 ± 7.85 | 161.9 ± 16.27 | 5.47 | 0.347 |
| 14 days | 273.31 ± 19.07 | 265.26 ± 19.72 | 285.96 ± 9.09 | 267.95 ± 35.09 | 8.47 | 0.422 |
| 21 days | 387.93 ± 30.39[a] | 385.19±17.84[a] | 430.88 ± 11.99[b] | 374.51±47.14[a] | 10.96 | 0.020 |
| 35 days | 842.09 ± 55.47[a] | 872.3 ± 39.12[bc] | 928.19±34.78[c] | 898.85 ± 47.99[bc] | 18.10 | 0.022 |
| ADG, g/day | | | | | | |
| 0–14 days | 12.94±1.22[a] | 13.11±0.92[a] | 14.89 ± 0.28[b] | 15.34 ± 0.09[b] | 0.26 | <0.001 |
| 21–35 days | 32±1.45[a] | 33.58 ± 1.48[ab] | 37.46±0.12[c] | 34.01 ± 1.61[b] | 0.48 | <0.001 |

[a]SEM, standard error of the mean ($n = 4$, number of replicates; larger sample sizes would enhance the reliability of the findings); BW, body weight; ADG, average daily gain.
[b]Different superscripts within a row indicate significant differences ($P < 0.05$).

**TABLE 8** Effect of compound microecological preparations on the intestinal morphology of young rabbits[a]

| Item | CON | Group A | Group B | Group C | SEM | P value |
|---|---|---|---|---|---|---|
| Jejunum | | | | | | |
| 14 days | | | | | | |
| Villus height, µm | 439.32 ± 126.88 | 473.64 ± 124.1 | 356.72 ± 45.16 | 408.14 ± 156.65 | 46.213 | 0.405 |
| Crypt depth, µm | 54.6 ± 8.93 | 57.66 ± 6.28 | 56.45 ± 10.18 | 53.16 ± 10.88 | 3.702 | 0.84 |
| Villus height/crypt depth | 8.03 ± 1.92 | 8.12 ± 1.48 | 6.5 ± 1.37 | 7.63 ± 2.5 | 0.743 | 0.432 |
| Intestinal wall thickness, µm | 59.48 ± 9.73 | 68.65 ± 11.25 | 77.34 ± 19.64 | 62.85 ± 10.01 | 5.167 | 0.136 |
| 35 days | | | | | | |
| Villus height, µm | 630.73 ± 110.95 | 591.96 ± 110.31 | 615.87 ± 99.55 | 648.38 ± 75.86 | 40.486 | 0.796 |
| Crypt depth, µm | 102.56 ± 18.45 | 103.38 ± 7.18 | 89.97 ± 13.11 | 104.72 ± 9.48 | 4.922 | 0.195 |
| Villus height/crypt depth | 6.23 ± 1.18 | 5.73 ± 0.99 | 6.9 ± 1.06 | 6.26 ± 1.09 | 0.441 | 0.344 |
| Intestinal wall thickness, µm | 128.21 ± 6.64 | 119.19 ± 8.22 | 131.4 ± 23.97 | 116.86 ± 19.33 | 7.271 | 0.55 |
| Ileum | | | | | | |
| 14 days | | | | | | |
| Villus height, µm | 245.98 ± 59.83 | 306.49 ± 81.59 | 341.12 ± 106.76 | 241.77 ± 64.71 | 31.936 | 0.124 |
| Crypt depth, µm | 61.56 ± 34.03 | 43.36 ± 10.84 | 48.75 ± 8.76 | 39.24 ± 5.1 | 5.995 | 0.212 |
| Villus height/crypt depth | 4.97 ± 2.38 | 7.07 ± 0.95 | 6.9 ± 1.07 | 6.21 ± 1.62 | 0.614 | 0.13 |
| Intestinal wall thickness, µm | 76.6 ± 21.18 | 62.36 ± 9.36 | 72.13 ± 7.65 | 76.89 ± 9.7 | 4.889 | 0.221 |
| 35 days | | | | | | |
| Villus height, µm | 487 ± 73.05 | 455.77 ± 54.69 | 520.19 ± 105.58 | 498.53 ± 87.3 | 32.723 | 0.599 |
| Crypt depth, µm | 99.8 ± 18.1 | 84.82 ± 9.73 | 84.3 ± 12.02 | 85.88 ± 15.37 | 5.636 | 0.21 |
| Villus height/crypt depth | 4.99 ± 1.03 | 5.44 ± 0.92 | 6.16 ± 0.85 | 5.82 ± 0.57 | 0.343 | 0.134 |
| Intestinal wall thickness, µm | 146.39 ± 22.24 | 145.33 ± 23.29 | 149.14 ± 33.2 | 137.61 ± 16.93 | 9.763 | 0.866 |
| Duodenum | | | | | | |
| 14 days | | | | | | |
| Villus height, µm | 445.48 ± 80.49 | 470.47 ± 92.54 | 462.34 ± 115.33 | 458.99 ± 127.92 | 42.486 | 0.981 |
| Crypt depth, µm | 54.62 ± 7.99 | 55.6 ± 3.44 | 59.73 ± 4.7 | 57.02 ± 10.26 | 2.693 | 0.633 |
| Villus height/crypt depth | 8.35 ± 2.31 | 8.46 ± 1.6 | 7.74 ± 1.92 | 8.09 ± 1.95 | 0.795 | 0.921 |
| Intestinal wall thickness, µm | 71.91 ± 11.38 | 76.75 ± 20.14 | 91.49 ± 24.8 | 77.61 ± 13.85 | 7.161 | 0.313 |
| 35 days | | | | | | |
| Villus height, µm | 906.66 ± 130.06 | 849.4 ± 148.95 | 895.4 ± 97.63 | 991.75 ± 71.95 | 45.784 | 0.227 |
| Crypt depth, µm | 97.44 ± 11.38 | 96.81 ± 9.93 | 102.53 ± 14.35 | 95.84 ± 9.2 | 4.579 | 0.745 |
| Villus height/crypt depth | 9.29 ± 0.76 | 8.82 ± 1.56 | 8.93 ± 1.89 | 10.44 ± 1.37 | 0.569 | 0.232 |
| Intestinal wall thickness, µm | 136.12 ± 28.08 | 142.09 ± 28.13 | 148.39 ± 32.45 | 132.17 ± 24.07 | 11.505 | 0.771 |

[a]SEM, standard error of the mean ($n = 4$, number of replicates; larger sample sizes would enhance the reliability of the findings).

## Effect on the intestinal microbiota of rabbit litters

### OTU statistics and diversity analysis

Based on the results presented in Figure 4 and Table 9, no significant impacts were observed on OTU number, Ace, diversity index (Chao1), and Shannon index (Shannon) in the cecal content samples of both 14-day-old and 35-day-old rabbits ($P > 0.05$). Similar to the female rabbits, the sequencing depth in the fresh fecal samples of young rabbits in all groups was sufficient, as indicated by a sequencing coverage value above 99% for the detected OTUs (Table 9).

According to the Shannon curve of microbial diversity index (Fig. 4A and B) constructed using varying sequencing depths of rabbit cecal content samples, it is evident that the sequence curves for each sample are relatively flat, indicating sufficient sequencing depth to accurately reflect most of the microbial information present in meat rabbit cecal contents and meet testing requirements.

### OTU Venn diagram analysis

At 14 days of age, a total of 1,355 OTUs were identified among the four groups, with an overlap of 435 OTUs as depicted in Figure 4C. Group C was the largest with 536 OTUs, of

**TABLE 9** Effect of compound microecological preparations on gut microbial diversity of young rabbits

| Item | CON | Group A | Group B | Group C | SEM[a] | P value |
|---|---|---|---|---|---|---|
| 14 days | | | | | | |
| OTU_num | 195.75 ± 31.67 | 187.75 ± 22.88 | 185.25 ± 14.31 | 197.25 ± 17.21 | 10.760 | 0.843 |
| Coverage | 99.99 ± 0.01 | 99.99 ± 0 | 99.99 ± 0.01 | 99.99 ± 0 | 0.002 | 0.644 |
| Ace | 197.39 ± 31.86 | 189.01 ± 23.02 | 186.35 ± 15.16 | 198.94 ± 17 | 10.879 | 0.827 |
| Chao1 | 196.56 ± 31.74 | 188.05 ± 22.98 | 185.94 ± 15.21 | 197.9 ± 16.89 | 10.852 | 0.839 |
| Shannon | 5.82 ± 0.34 | 5.89 ± 0.39 | 5.62 ± 0.2 | 5.97 ± 0.24 | 0.147 | 0.444 |
| 35 days | | | | | | |
| OTU_num | 453.25 ± 37.62 | 503.75 ± 72.41 | 481.75 ± 43.42 | 484.5 ± 46.34 | 24.973 | 0.599 |
| Coverage | 99.97 ± 0.01 | 99.97 ± 0.01 | 99.97 ± 0.01 | 99.97 ± 0.02 | 0.005 | 0.912 |
| Ace | 460.1 ± 38.11 | 509.09 ± 72.68 | 485.75 ± 43.63 | 491.81 ± 48.34 | 25.345 | 0.627 |
| Chao1 | 456.3 ± 38.05 | 505.54 ± 72.12 | 482.8 ± 43.38 | 488.52 ± 47.65 | 25.148 | 0.617 |
| Shannon | 7.14 ± 0.18 | 7.27 ± 0.27 | 7.27 ± 0.28 | 7.36 ± 0.18 | 0.114 | 0.640 |

[a]SEM, standard error of the mean ($n = 4$, number of replicates; larger sample sizes would enhance the reliability of the findings).

which 242 were unique OTUs. Group B had the fewest, with 501 OTUs, of which 194 were unique. Moving on to the 35-day-old rabbits, a total of 4,499 OTUs were observed among the four groups, with 1,068 overlapping. Group A was the largest with 1,618 OTUs, of which 900 were unique OTUs. The control group had the fewest, with 1,477 OTUs, of which 779 were unique (Fig. 4D).

## NMDS and PERMANOVA

Similar to the female rabbits, NMDS and PERMANOVA based on Bray-Curtis distance showed that the stress function value of the community structure of the intestinal flora of 14-day-old rabbits was 0.14. Compared with the control group, the experimental groups were mainly distributed in the negative range of NMDS1 (Fig. 4E), indicating that the structure of the intestinal flora of the experimental groups and the control group was significantly different when the young rabbits were fed with breast milk ($P = 0.017$) (Fig. 4G). As shown in Figure 4F, the value of the stress function for the structure of the intestinal flora in 35-day-old rabbits was 0.21, and the distribution was relatively dispersed with no significant difference observed between the four groups at both time periods ($P > 0.05$) (Fig. 4H).

## Analysis of cecal microbial community structure at the phylum level

It can be seen from Figure 5A and Table 10 that at the level of phylum, the cecal flora of each experimental group primarily consisted of *Firmicutes*, *Bacteroidetes*, and *Verrucomicrobia* at 14 days of age. There were no significant differences between the control group and the experimental groups of *Firmicutes*, *Bacteroidetes*, and a *Verrucomicrobia* ($P > 0.05$). Similarly, at 35 days of age (Fig. 5B; Table 10), the cecal flora of each experimental group was also mainly composed of *Firmicutes*, *Bacteroidetes*, and a *Verrucomicrobia*. Specifically, the abundance of *Firmicutes* in group C was higher compared with other groups; meanwhile, the abundance of *Verrucomicrobiota* in groups A and B was higher than that in the control group.

## Analysis of cecal microbial community structure at the family level

From Figure 5C and D and Table 10, it is apparent that at the family level, there are differences in the abundance of dominant flora between the two time periods, whereas no significant differences were observed among the groups at both 14 and 35 days of age ($P > 0.05$). At 14 days of age, compared with the control group, the abundance of *Bacteroides* in the experimental groups had a decreasing trend. At 35 days of age, differences were observed between the control group and the experimental groups A

**TABLE 10** Relative abundance of OTUs about the cecal microflora of young rabbits[a,]

| Item | CON | Group A | Group B | Group C | SEM | *P* value |
|---|---|---|---|---|---|---|
| Phylum | | | | | | |
| 14 days | | | | | | |
| *Bacteroidota* | 43.54 ± 10.08 | 37.42 ± 5.1 | 37.69 ± 5.15 | 35.85 ± 5.75 | 3.259 | 0.437 |
| *Firmicutes* | 41.03 ± 7.07 | 35.98 ± 9.12 | 37.04 ± 11.32 | 34.61 ± 6.06 | 4.197 | 0.75 |
| *Verrucomicrobiota* | 6.04 ± 4.34 | 6.22 ± 1.07 | 6.63 ± 2.6 | 6.72 ± 0.78 | 1.098 | 0.979 |
| 35 days | | | | | | |
| *Bacteroidota* | 25.19 ± 3.69 | 23.39 ± 1.38 | 23.49 ± 2.83 | 21.7 ± 1.54 | 1.181 | 0.333 |
| *Firmicutes* | 65.89 ± 4.29 | 65.69 ± 1.11 | 63.77 ± 1.97 | 69.09 ± 2.2 | 1.197 | 0.09 |
| *Verrucomicrobiota* | 4.08 ± 2.4 | 4.54 ± 2.76 | 6.83 ± 0.63 | 3.19 ± 2.35 | 1.018 | 0.168 |
| Family | | | | | | |
| 14 days | | | | | | |
| *Bacteroidaceae* | 25.54 ± 6.14 | 17.88 ± 5.62 | 21.26 ± 4.15 | 15.93 ± 4.41 | 2.54 | 0.094 |
| *Lachnospiraceae* | 21.31 ± 4.96 | 20.14 ± 5.56 | 18.38 ± 5.74 | 16.12 ± 2.65 | 2.364 | 0.49 |
| *Rikenellaceae* | 6.18 ± 5.41 | 11.69 ± 9.19 | 9.29 ± 4.75 | 12.47 ± 1.56 | 2.613 | 0.459 |
| *Oscillospiraceae* | 8 ± 2.54 | 7.74 ± 2.37 | 10.48 ± 7.18 | 10.71 ± 3.33 | 1.928 | 0.667 |
| 35 days | | | | | | |
| *Lachnospiraceae* | 17.61 ± 5.17 | 18.16 ± 3.8 | 17.09 ± 3.2 | 21.49 ± 2.49 | 1.832 | 0.39 |
| *Oscillospiraceae* | 17.2 ± 2.42 | 16.14 ± 2.58 | 14.54 ± 2.52 | 15.24 ± 2.43 | 1.244 | 0.491 |
| *Ruminococcaceae* | 12 ± 3.95 | 10.12 ± 1.93 | 11.3 ± 2.94 | 10.85 ± 3.66 | 1.561 | 0.865 |
| *Bacteroidaceae* | 10.99 ± 2.88 | 10.35 ± 4.91 | 7.25 ± 3.82 | 9.57 ± 3.54 | 1.893 | 0.56 |
| *Akkermansiaceae* | 0.04 ± 0.02 | 0.05 ± 0.03 | 0.07 ± 0.01 | 0.03 ± 0.02 | 0.010 | 0.168 |
| Genus | | | | | | |
| 14 days | | | | | | |
| *Bacteroides* | 25.54 ± 6.14 | 17.88 ± 5.62 | 21.26 ± 4.15 | 15.93 ± 4.41 | 2.54 | 0.094 |
| UL | 11.92 ± 5.88 | 18.28 ± 4.1 | 16.56 ± 5.69 | 12.4 ± 1.48 | 2.144 | 0.198 |
| *Akkermansia* | 6.04 ± 4.34 | 6.22 ± 1.07 | 6.63 ± 2.6 | 6.72 ± 0.78 | 1.098 | 0.979 |
| *NK4A214_group* | 3.58 ± 1.7 | 3.92 ± 1.59 | 6.98 ± 6.55 | 6.01 ± 2.86 | 1.586 | 0.536 |
| 35 days | | | | | | |
| UL | 14 ± 5.86 | 13.57 ± 2.26 | 14.55 ± 2.66 | 18.03 ± 2.25 | 1.628 | 0.324 |
| *Bacteroides* | 10.99 ± 2.88 | 10.35 ± 4.91 | 7.25 ± 3.82 | 9.57 ± 3.54 | 1.893 | 0.56 |
| *NK4A214_group* | 6.43 ± 1.16 | 6.22 ± 1.14 | 4.7 ± 0.42 | 6.18 ± 1.33 | 0.507 | 0.141 |
| UCV_group | 3.68 ± 2.28 | 5.07 ± 1.64 | 6.32 ± 3.9 | 6.45 ± 3.74 | 1.445 | 0.559 |

[a]SEM, standard error of the mean (*n* = 4, number of replicates; larger sample sizes would enhance the reliability of the findings); UL, *unclassified_Lachnospiraceae*; UCV, *unclassified_Clostridia_vadinBB60_group*.

and B regarding *Akkermansiaceae*, among them, the abundance of experimental group B was the highest.

### *Analysis of gut microbial community structure at the genus level*

As observed in Figure 5E and F and Table 10, at the genus level, similar to the family level, there were differences in the abundance of dominant bacteria between the two time periods. There was no significant difference among the groups during both the 14-day-old and 35-day-old periods (*P* > 0.05). At 14 days of age, with the intake of compound microecological preparations, the abundance of *Akkermansiaceae* showed an increasing trend.

## DISCUSSION

### Lactation performance and growth performance

The lactation performance of female rabbits plays a pivotal role in determining the growth and development of young rabbits. Therefore, it holds significant production implications to investigate the impact of supplementary compound probiotics on the

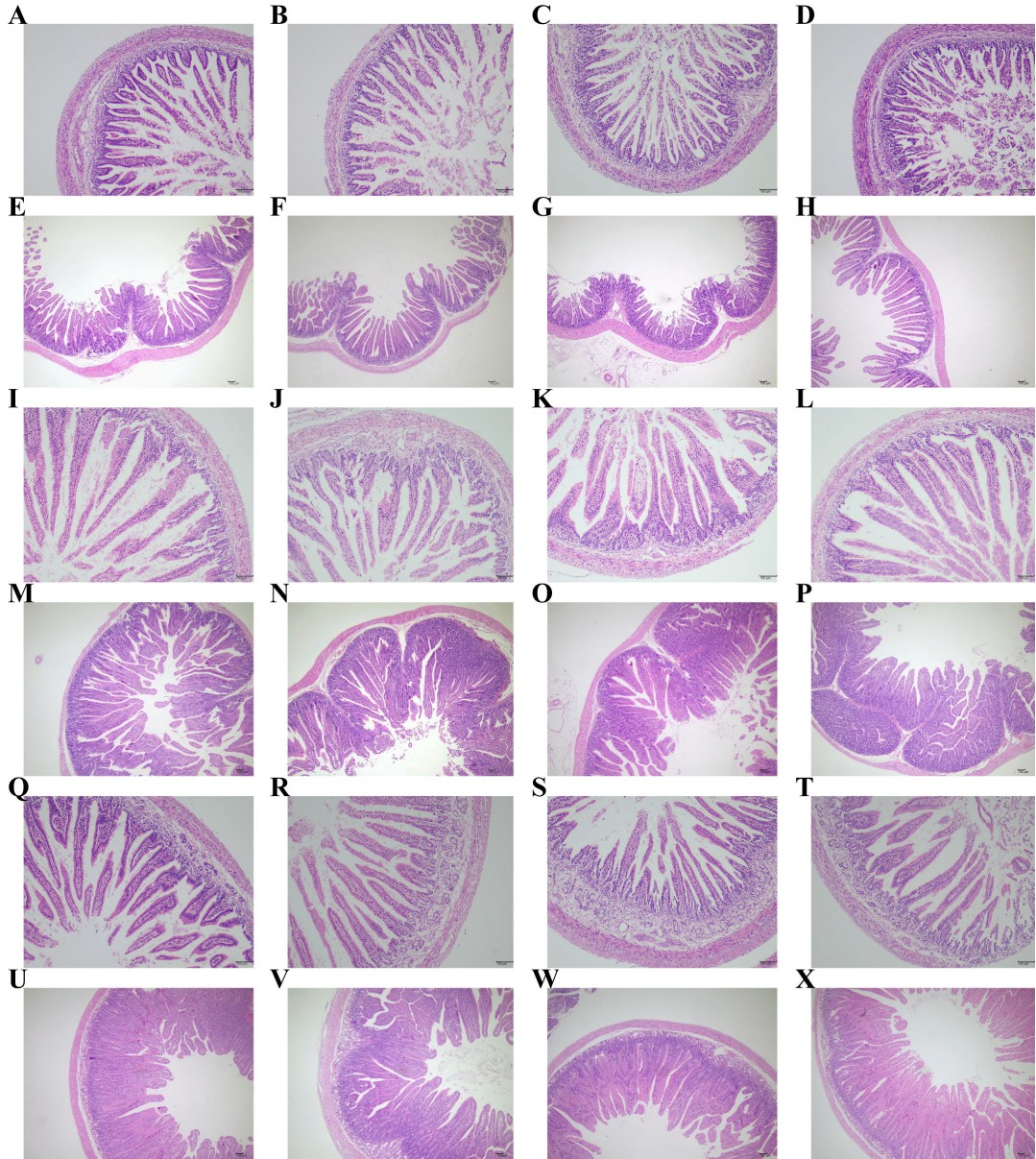

**FIG 3** Effect of compound microecological preparations on the intestinal morphology of young rabbits. Ileum of newborn rabbits in (A) control group, (B) group A, (C) group B, and (D) group C on day 14. Ileum of newborn rabbits in (E) control group, (F) group A, (G) group B, and (H)Ileum of newborn rabbits in group C on day 35. Jejunum of newborn rabbits in (I) control group, (J) group A, (K) group B, and (L) group C on day 14. Jejunum of newborn rabbits in (M) control group, (N) group A, (O) group B, and (P) group C on day 35. Duodenum of newborn rabbits in (Q) control group, (R) group A, (S) group B, and (T) group C on day 14. Duodenum of newborn rabbits in (U) control group, (V) group A, (W) group B, and (X) group C on day 35.

lactation performance of female rabbits. Numerous studies have demonstrated that supplementation with compound microecological preparations exerts a significant positive impact on both milk yield and quality in dairy cattle, goats, sows, and other livestock (49–51). This study demonstrated that the milk yield of the female rabbits in the experimental group exhibited a significant increase of 19.23%, 44.22%, and 24.57% compared with the control group ($P = 0.002$). After evaluating the birth weight, as well as the weights at 7, 14, 21, and 35 days of age in young rabbits, it was observed that the supplementation of compound microecological preparation resulted in an increase in rabbit weight from 21 days until the weaning stage, and the average body weight of rabbits in the experimental groups A, B, and C increased significantly by 3.59%, 10.22%,

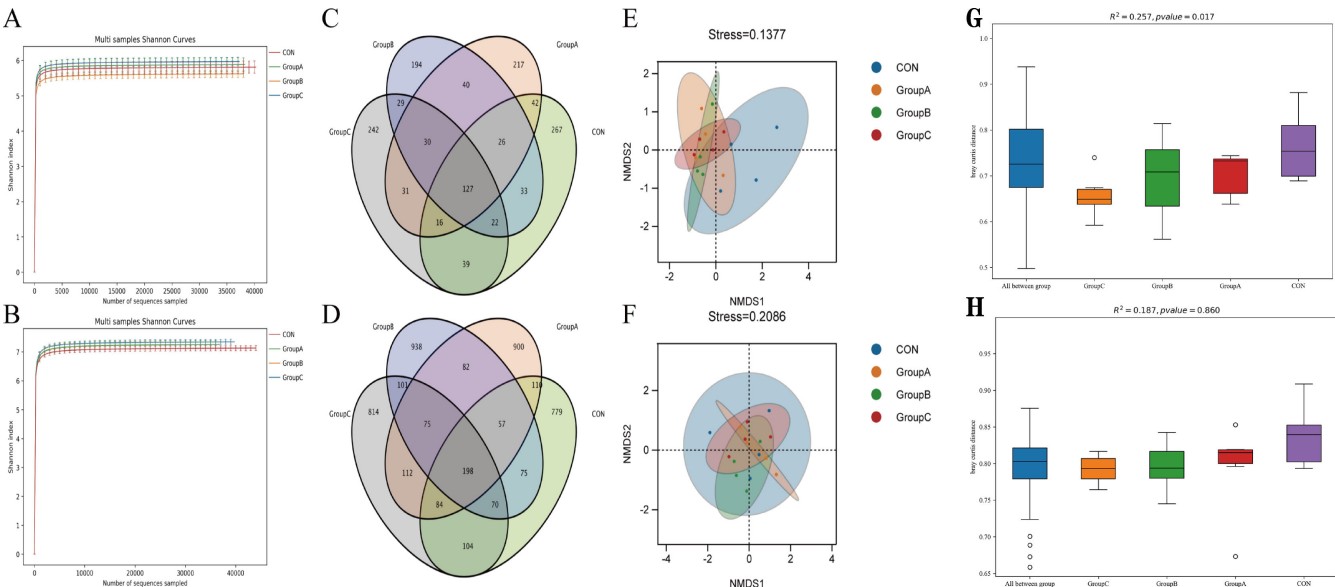

**FIG 4** Effect on the intestinal microbiota of rabbit litters. (A) Shannon curve, (C) OTU Venn diagram, (E) NMDS, and (G) PERMANOVA of 14-day-old rabbits. (B) Shannon curve, (D) OTU Venn diagram, (F) NMDS, and (H) PERMANOVA of 35-day-old rabbits.

**FIG 5** Effect on the intestinal microbiota of rabbit litters. Relative abundance of OTUs about the cecal microflora of 14-day-old rabbits at the (A) phylum, (C) family, and (E) genus levels. Relative abundance of OTUs about the cecal microflora of 35-day-old rabbits at the (B) phylum, (D) family, and (F) genus levels.

and 6.74% ($P = 0.022$) on the 35th day compared with the control group. The average daily gains from 21 to 35 days were assessed and found to exhibit significant increases of 4.94%, 17.06%, and 6.28% ($P < 0.001$) in the experimental groups, respectively. These findings indirectly suggest that incorporating compound microecological preparations into the diet can enhance lactation performance of rabbits during late gestation and lactation. Furthermore, regarding the promotion of lactation-related hormones, it was observed that the E2 levels in the experimental group were significantly reduced by 13.52%, 31.27%, and 18.32% compared with the control group ($P < 0.001$), whereas GH levels showed a significant increase of 12.31%, 22.82%, and 26.41% ($P < 0.05$). Therefore, we hypothesize that the compound probiotics may potentially enhance the lactation performance in female rabbits by employing two distinct mechanisms. One is that E2 and GH play pivotal roles in enhancing the lactation yield of female rabbits. The elevation of plasma E2 levels is often accompanied by a reduction in lactation yield (52). GH exerts a positive influence on mammary gland development (53); therefore, the compound microecological preparation can reinforce mammary gland development to enhance lactation yield by inhibiting the activation and release of plasma E2 and promoting GH secretion. The other is that the compound microecological preparation possesses a biological function of augmenting the body's immune response (Table 3). This enhancement in immunity diminishes the likelihood of diseases such as mastitis, thereby significantly ameliorating milk production in female rabbits.

However, our study failed to demonstrate a significant improvement in milk quality through the supplementation of compound microecological preparation. There were no notable disparities observed in milk fat, lactose, and milk protein contents between the experimental group and the control group (Table 2). The variations in the outcomes may be attributed to the dietary composition, probiotic supplementation methodology and dosage, as well as the physiological state of the experimental subjects. Furthermore, Gilman et al. (54) demonstrated that PRL exerts a regulatory effect on lactose content reduction while simultaneously enhancing milk fat and milk protein levels, whereas our finding demonstrated that the supplementation of compound microecological preparations did not yield a statistically significant impact on plasma PRL concentration. In summary, our study provides evidence that the compound microecological preparation exhibits potential in enhancing lactation performance in lactating female rabbits, although further investigation is warranted to elucidate its underlying mechanism.

## Plasma biochemical markers

Blood plays a crucial role in the internal milieu by facilitating the transportation of diverse metabolites throughout the animal body. Biochemical markers can provide insight into the body's metabolic processes to a certain extent (55, 56). Hepatic enzymes such as ALT and LDH are released into the bloodstream when the liver is damaged, resulting in elevated plasma levels (57). Consequently, plasma ALT and LDH concentrations can serve as indicators of liver function (58). When the content of UN in plasma decreases, there is an increase in nitrogen precipitation within the body and a more efficient utilization of dietary protein. Conversely, an increase in UN content indicates a decrease in protein utilization rate and an increase in protein catabolism (59). Plasma TG levels serve as a crucial indicator for assessing blood lipid profiles, with deviations from normal ranges serving as early warning signs of potential disease onset (60). In this study, no significant differences were observed in plasma biochemical indexes between the experimental and control groups of lactating female and young rabbits, indicating that the supplementation of compound probiotics did not lead to an increase in the body burden of experimental rabbits and had no discernible impact on their overall health.

## Immune and antioxidant indexes

Antioxidant capacity and immune function are the two most important indicators to measure the health status and disease resistance of animals. The enzymatic antioxidant system is composed of SOD, GSH-PX, catalase, and other enzymes (61). It is widely

acknowledged that CAT and SOD, two crucial endogenous antioxidant enzymes, play pivotal roles in preventing oxidative damage (62). MDA, a byproduct of cell membrane peroxidation, serves as an indirect indicator of cellular damage, specifically lipid damage resulting from oxidative stress caused by free radicals (63, 64). Zhang and Li (65) discovered that the intake of composite microecological preparations significantly enhances the activity level of GSH-PX in the serum of meat rabbits, with GSH-PX serving as the body's primary antioxidant defense mechanism. Previous studies have also demonstrated that dietary supplementation with *Bacillus amyloliquefaciens 40* can enhance the antioxidant capacity by increasing total antioxidant capacity (T-AOC) and reducing MDA concentrations in plasma among dairy sows (66, 67), indicating that the compound microecological preparation effectively enhances the body's antioxidant capacity, aligning with the outcomes of this experimental study.

Immunoglobulins, including IgG, IgA, and IgM, play pivotal roles in the body's immune system, and their quantities are positively correlated with the immune competence of an individual (68). The sIgA is the most prevalent colonic antibody antigen known as "immune exclusion," which has the potential to enhance animal immune status (69). Cytokines, as small proteins, play a crucial role in regulating immune responses by interacting with specific receptor cells (70, 71). Our findings demonstrate that dietary supplementation of compound microecological preparations during lactation significantly enhances the levels of immunoglobulin (IgG, IgA, IgM), sIgA, and anti-inflammatory factors (IL-2, IL-8) in plasma while suppressing the release of pro-inflammatory factors (IL-6, TNF-α). Conversely, indirect administration of compound microecological preparations through rabbit milk solely impacts certain immune and antioxidant markers in young rabbits, potentially attributable to inadequate intake during lactation.

## Intestinal microbiota and intestinal morphology

The intestinal microbiota represents the largest and most intricate microecological system in animals, colonizing their bodies from birth and persisting throughout their lifespan (72–74). The compound microecological preparations utilized in this study consist of three distinct microecological formulations: *B. subtilis*, *B. licheniformis*, and *S. cerevisiae*. Both *Firmicutes* and *Bacteroidetes* play a crucial role in the rabbit intestine, with *Firmicutes* demonstrating greater stability compared with *Bacteroidetes*. Consequently, an increased ratio of *Firmicutes* to *Bacteroidetes* contributes to enhanced protection against harmful bacteria, thereby promoting intestinal health (75). In the female rabbits, the abundance of *Firmicutes* was found to be higher in group B than in the control group. Similarly, analysis of the intestinal microbiota in the young rabbits revealed a higher abundance of *Firmicutes* in the experimental group compared with the control group, aligning with previous research findings. The findings of this study further demonstrate that the administration of compound microecological preparations to lactating female rabbits can significantly enhance the abundance of *Rikenellaceae* in their gut microbiota. *Rikeneceae* are frequently observed in the gastrointestinal tract of mammals, and their abundance is closely associated with dietary patterns (76). A previous study has indicated that the *Rikeneceae_RC9_gut_group* may play a crucial role in the breakdown of crude fiber, whereas the *Rikenella_RC9_gut_group* genus is essential for feed absorption and utilization in Tan sheep (77). Therefore, we hypothesize that *Rikenella_RC9_gut_group* enhances the tricarboxylic acid cycle activity in female rabbits by accelerating glucose production through dietary cellulose degradation. This mechanism ensures sufficient energy generation to meet the body's requirements and reduces reliance on external energy sources. Unfortunately, this study did not investigate the metabolic pathways of tricarboxylic acid in breast tissue, thus necessitating further research.

A previous study has highlighted the significance of *Akkermansia*, an anaerobic Gram-negative bacterium, in the gastrointestinal tract (78). Our findings revealed a tendency for higher levels of *Akkermansiaceae* and *Akkermansia* in the experimental

group compared with the control group in the young rabbits, with Group B exhibiting significantly augmented abundance relative to other groups. *Akkermansia muciniphila*, as a representative strain of *Verrucomicrobia*, constitutes over 80% of the total *Verrucomicrobia* population, which is related to intestinal barrier function, immune response, and host metabolism (79, 80). The bacteria predominantly colonized the outer mucus layer of the terminal ileum, colon, and cecum, constituting approximately 1% to 4% of the total intestinal bacterial population (81). The findings of multiple studies have consistently demonstrated a significant reduction in the abundance of *A. muciniphila* in patients with obesity (82), diabetes, metabolic disorders (83), and inflammatory bowel disease, particularly ulcerative colitis (84). These findings suggest that *A. muciniphila* may play a potential role in modulating host immune function, exerting anti-bacterial and anti-inflammatory effects, as well as preventing enteropathogenic bacterial infections. Due to the limitations of the experimental design, we did not quantify the specific abundance of *A. muciniphila* in each group. Further investigations are warranted to explore the impact of supplementing complex microecological preparations on gut colonization by *A. muciniphila*.

In this study, intestinal samples from the jejunum, ileum, and duodenum were collected to investigate the impact of compound microecological preparations on young rabbit intestine morphology. Gut morphology is considered a critical parameter for assessing gut health and injury (85). Villus height and crypt depth reflect the digestive and absorptive capacity of intestinal epithelial cells (86), and increasing the surface area of intestinal mucosa can enhance nutrient digestion and absorption (87, 88). The results of this study indicate that the V/C value of ileum in the experimental group exhibited a higher trend than that in the control group, suggesting that probiotic supplementation exerted beneficial effects on ileal in the young rabbits and facilitated intestinal development.

## Conclusion

Our findings highlight the potential of compound microecological preparations as an effective strategy for enhancing lactation performance, immune function, and antioxidant capacity in rabbits. The supplementation of probiotics through RBM offers a promising approach to optimize the growth and health outcomes of newborn rabbits. Further research is required to elucidate the underlying mechanisms and explore the long-term effects of compound microecological preparations on rabbit health and production performance. In summary, this study suggests that the inclusion of compound microecological preparations at a dosage of 6 g/day in the diets of lactating rabbits can effectively enhance both performance and health status of lactating rabbits and their offspring.

Additionally, the practical implications of our study extend to the development of more targeted probiotic formulations, enhancing animal welfare and productivity in sustainable farming practices. Future research should focus on longitudinal studies to ascertain the long-term impacts of such interventions, including potential generational effects.

## ACKNOWLEDGMENTS

This study was supported by the projects of Research and Development and Demonstration and Promotion of Key Technologies for Scale and Standardized Healthy Breeding of Rabbits (NYKJ-2020-YL-16v), Integration and Promotion of Technological Innovation for Improving Quality and Efficiency in the Rabbit Industry [NYKJ-2021-YL(XN)28v], and Research and Promotion of Integrated Nutritional Technology for Meat Rabbits and Mothers and Litters [NYKJ-2021-YL(XN)26J].

C.Z.: conceptualization, data curation, investigation, methodology, interpretation of the statistical outcome, writing—original draft, writing—review and editing; Y.L., H.W., and A.I.S.: conceptualization, funding acquisition, methodology, resources, supervision,

formal analysis; S.W., X.D., and B.S.: conceptualization, supervision; Z.R.: supervision, writing—review and editing.

We declare that we have no financial and personal relationships with other people or organizations that can inappropriately influence our work, and there is no professional or other personal interest of any nature or kind in any product, service, and/or company that could be construed as influencing the content of this paper.

## AUTHOR AFFILIATION

[1]College of Animal Science and Technology, Northwest A&F University, Yangling, Shaanxi, China

## AUTHOR ORCIDs

Chengcheng Zhao ⓘ http://orcid.org/0009-0003-7722-714X
Zhanjun Ren ⓘ http://orcid.org/0000-0001-5607-172X

## ETHICS APPROVAL

All animal experiments were approved by the Executive Committee of Laboratory Animal Management and Ethical Review of Northwest Agriculture and Forestry University of Science and Technology in Shaanxi Province.

## ADDITIONAL FILES

The following material is available online.

Open Peer Review

**PEER REVIEW HISTORY (review-history.pdf).** An accounting of the reviewer comments and feedback.

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
