## [Reviewer comments · Microbiology Spectrum]

Microbiology Spectrum

Dietary supplementation with compound microecological preparations: Effects on the production performance and gut microbiota of lactating female rabbits and their litters

Chengcheng Zhao, Youhao Li, Hui Wang, Ahamba Ifeanyi Solomon, Shuhui Wang, Xianggui Dong, Bing Song, and Zhanjun Ren

Corresponding Author(s): Zhanjun Ren, College of Animal Science and Technology, Northwest A&F University

Review Timeline:

Submission Date:	January 6, 2024
Editorial Decision:	March 2, 2024
Revision Received:	April 11, 2024
Accepted:	August 4, 2024

Editor: Francisco Uzal

Reviewer(s): The reviewers have opted to remain anonymous.

Transaction Report:

DOI: <https://doi.org/10.1128/spectrum.00067-24>

Re: Spectrum00067-24 (Dietary supplementation with compound microecological preparations: Effects on the production performance and gut microbiota of lactating female rabbits and their litters)

Dear Prof. Zhanjun Ren:

Thank you for the privilege of reviewing your work. Below you will find my comments, instructions from the Spectrum editorial office, and the reviewer comments.

Revision Guidelines

Sincerely,
Francisco Uzal
Editor
Microbiology Spectrum

Reviewer #1 (Comments for the Author):

Thank you for providing the opportunity to offer feedback on the research article titled "Dietary supplementation with compound microecological preparations: Effects on the production performance and gut microbiota of lactating female rabbits and their litters". The authors investigate the effects of compound microecological preparations on lactating rabbits and their offspring. Their results suggest that a dose of 6g/day shows the most beneficial effects, enhancing lactation performance,

immune

function, and antioxidant capacity. However, the presentation of descriptions lacks precision and clarity, resulting in ambiguity for readers.

It is imperative that the authors address these comments to enhance the manuscript's quality. Section wise comments are listed as;

Abstract:

1. While the abstract mentions improvements in lactation performance, immune performance, antioxidant performance, and growth performance, it does not provide specific quantitative results or effect sizes for these improvements. Readers may benefit from more precise numerical data to understand the magnitude of the effects observed.

2. The abstract does not mention any potential limitations of the study, such as sample size, experimental design. Including a brief description of these aspects could provide readers with a more comprehensive understanding of the study's findings and their implications.

3. Introduction:

I have noticed following shortcomings in introduction section;

1. The introduction primarily cites older references, and there is a lack of recent studies to support the discussion. Including more up-to-date references could strengthen the relevance and validity of the arguments presented (Lines 43, 44).

2. While the introduction mentions the potential benefits of various probiotics and compound microecological preparations, it does not provide specific details about the strains used in this study. Providing information about the specific strains and their known effects could enhance the clarity and specificity of the introduction (Lines 40-42).

3. The introduction outlines the importance of finding alternatives to antibiotics for improving rabbit health but does not clearly articulate the specific research gap or the novelty of the current study. Including a clearer statement about the gap in the literature that this study aims to address could provide readers with a better understanding of its significance.

Material and methods:

Please address these issues in material and methods section

1. While the study mentions that animals were randomly assigned to groups, it does not provide specific details about the randomization process. Providing information about how randomization was conducted could enhance the transparency and reproducibility of the study

2. The section describing sample collection for both young rabbits and female rabbits lacks detailed information about standardized procedures followed during sample collection. Providing specific details about standardized protocols for sample collection could ensure consistency and reliability of the data.

3. Although the statistical analysis is mentioned, the section lacks specific details about the types of statistical tests used and how the data were analyzed. Providing more information about the statistical methods employed could help readers better understand the robustness of the study's findings.

Results

1. The results do not include comparisons with baseline values or references to normal ranges for the measured parameters in rabbits. This makes it challenging to interpret the significance of the observed changes without understanding the starting point.

2. While statistical tests are mentioned for some comparisons, it's unclear whether adjustments were made for multiple comparisons. Without proper correction, there's a risk of false positive results, especially when analyzing multiple biochemical markers and immune indexes.

3. The sample size and number of replicates are mentioned ($n=4$), which may be considered low for drawing robust conclusions, particularly in complex biological systems like the gut microbiota. So authors could a sentence for this potential shortcomings that "Larger sample sizes would enhance the reliability of the findings".

4. The results do not mention whether potential confounding factors, such as age, sex, or health status of the rabbits, were controlled for in the analysis. Failure to account for these variables could introduce bias into the results.

Discussion

1. The discussion predominantly focuses on summarizing the results and existing literature without providing a critical analysis or interpretation of the findings. A more in-depth discussion that critically evaluates the implications of the results in the context of the research field could enhance the manuscript's quality.

2. While the discussion briefly mentions previous studies that support the findings, there is a lack of detailed comparison with existing literature. Providing a more comprehensive comparison with previous research, including both similarities and differences, could strengthen the discussion and highlight the novelty of the current study.

3. Some points in the discussion are repeated, leading to redundancy and lack of conciseness. Streamlining the discussion and avoiding repetition would improve clarity and readability.

Conclusion

The conclusion fails to provide a clear recommendation or actionable insight based on the study's findings. While it mentions the potential benefits of compound microecological preparations, it does not offer specific guidance or practical implications for future research or application in rabbit husbandry practices.

Please answer to these question positively

1. What were the effects of compound microecological preparations on the plasma biochemical markers of rabbits?

2. Were there any significant differences observed in the biochemical indicators between the experimental and control groups of female rabbits?

3. Can you provide information on the immune and antioxidant indexes of rabbits after the administration of compound microecological preparations?

4. What were the findings regarding the intestinal microbiota of female rabbits after the administration of compound microecological preparations ?

Reply to the Review Comments

Thank you for your patient and meticulous revision of the article. Your good comments have helped us a lot. The following is our reply to your comments.

Reviewer #1 (Comments for the Author):

Thank you for providing the opportunity to offer feedback on the research article titled "Dietary supplementation with compound microecological preparations: Effects on the production performance and gut microbiota of lactating female rabbits and their litters". The authors investigate the effects of compound microecological preparations on lactating rabbits and their offspring. Their results suggest that a dose of 6g/day shows the most beneficial effects, enhancing lactation performance, immune function, and antioxidant capacity. However, the presentation of descriptions lacks precision and clarity, resulting in ambiguity for readers.

It is imperative that the authors address these comments to enhance the manuscript's quality. Section wise comments are listed as;

Abstract:

1. While the abstract mentions improvements in lactation performance, immune performance, antioxidant performance, and growth performance, it does not provide specific quantitative results or effect sizes for these improvements. Readers may benefit from more precise numerical data to understand the magnitude of the effects observed.

- Thank you so much for your valuable suggestions regarding the abstract section of this article. We have edited the abstract to focus on the effects of complex microecological preparations on the lactation performance of female rabbits and the growth performance of young rabbits. And under your guidance and suggestions, we have revised the presentation of our results and incorporated more accurate quantitative improvements to ensure a clearer representation of our experimental findings.

“Early weaning is frequently accompanied by a significant increase in diarrhea and mortality rates, which reduces the rabbits' performance. While antibiotics can reduce pathogenic bacteria, they also harm beneficial microorganisms and disrupt the normal intestinal microbiota balance. In order to find non-residue and non-toxic alternatives to antibiotics to ensure the safety of animal products, We conducted a study on the effect of compound microecological preparations supplementation on lactating female rabbits and their offspring. A total of 60 female rabbits were randomly assigned to four groups: CON, supplemented with probiotics at 3, 6, and 9 g/female rabbit per day from day 24 of gestation until weaning. We observed that probiotics supplementation significantly enhanced production performance ($P<0.05$), immune and antioxidant function ($P<0.05$), as well as intestinal flora composition in lactating rabbits and their offspring. Notably, compared to the control group, the experimental group exhibited a 19.23%, 44.22%, and 24.57% increase in milk yield ($P=0.002$). Regarding rabbit growth performance, the average body weight of young rabbits in the experimental group showed a significant increase of 3.59%, 10.22%, and 6.74% at day 35 ($P=0.022$), while the average daily gain of rabbits aged between 21-35 days was significantly elevated by 4.94%, 17.06%, and 6.28% in the experimental

group ($P<0.001$). In conclusion, probiotics supplementation can significantly enhance lactation performance, promote growth and disease resistance in rabbits, as well as improve intestinal health when administered at a dosage of 6g/d. Moreover, the limited sample size in this study may hinder the detection of subtle effects, and augmenting the sample size will bolster the reliability of the study findings.”

2. The abstract does not mention any potential limitations of the study, such as sample size, experimental design. Including a brief description of these aspects could provide readers with a more comprehensive understanding of the study's findings and their implications.

- Thank you so much for your valuable suggestions. We have now added the limitations mentioned in the abstract section.

“Early weaning is frequently accompanied by a significant increase in diarrhea and mortality rates, which reduces the rabbits' performance. While antibiotics can reduce pathogenic bacteria, they also harm beneficial microorganisms and disrupt the normal intestinal microbiota balance. In order to find non-residue and non-toxic alternatives to antibiotics to ensure the safety of animal products, We conducted a study on the effect of compound microecological preparations supplementation on lactating female rabbits and their offspring. A total of 60 female rabbits were randomly assigned to four groups: CON, supplemented with probiotics at 3, 6, and 9 g/female rabbit per day from day 24 of gestation until weaning. We observed that probiotics supplementation significantly enhanced production performance ($P<0.05$), immune and antioxidant function ($P<0.05$), as well as intestinal flora composition in lactating rabbits and their offspring. Notably, compared to the control group, the experimental group exhibited a 19.23%, 44.22%, and 24.57% increase in milk yield ($P=0.002$). Regarding rabbit growth performance, the average body weight of young rabbits in the experimental group showed a significant increase of 3.59%, 10.22%, and 6.74% at day 35 ($P=0.022$), while the average daily gain of rabbits aged between 21-35 days was significantly elevated by 4.94%, 17.06%, and 6.28% in the experimental group ($P<0.001$). In conclusion, probiotics supplementation can significantly enhance lactation performance, promote growth and disease resistance in rabbits, as well as improve intestinal health when administered at a dosage of 6g/d. Moreover, the limited sample size in this study may hinder the detection of subtle effects, and augmenting the sample size will bolster the reliability of the study findings.”

Introduction:

I have noticed following shortcomings in introduction section;

1. The introduction primarily cites older references, and there is a lack of recent studies to support the discussion. Including more up-to-date references could strengthen the relevance and validity of the arguments presented (Lines 43, 44).

- According to your suggestion, we have thoroughly examined the recent research advancements in this field and successfully completed the necessary modifications for this section.

“With the expansion of large-scale intensive farming, the prevalence of intestinal diseases has emerged as a primary constraint on the growth of the rabbit industry (Mancini et al. 2021). ”

2. While the introduction mentions the potential benefits of various probiotics and compound microecological preparations, it does not provide specific details about the strains used in this study. Providing information about the specific strains and their known effects could enhance the clarity and specificity of the introduction(Lines 40-42).

- Thank you so much for your valuable guidance and suggestions. Through a thorough literature review, we have gained a comprehensive understanding of the functions and effectiveness of probiotics used in this study. We have then incorporated these findings into the introduction section of this article with detailed explanations.

“*Bacillus subtilis* has the ability to secrete surfactants that possess antibacterial activity, thereby enhancing intestinal antioxidant status, fortifying the stability of gut microbiota and regulating its composition in laying hens for maintaining optimal intestinal health (Zou et al. 2022). Bortoluzzi et al. discovered that the regulation of intestinal microbiota diversity and composition by *B. subtilis* can effectively mitigate necrotizing enteritis lesions and growth performance decline induced by *C. perfringens* (Bortoluzzi et al. 2019). *Bacillus licheniformis* is a bacterium of significant commercial value due to its ability to synthesize a diverse range of bacteriocins, antimicrobial peptides, and digestive enzymes, and also these bioactive compounds can effectively inhibit pathogenic microorganisms, enhance the stability of intestinal flora, optimize feed nutritional composition, and facilitate efficient digestion and absorption processes within the host organism (Zhao et al. 2020). The findings of various studies have demonstrated that dietary supplementation with *Bacillus licheniformis* can effectively enhance growth performance, improve heat stress tolerance, and exert preventive effects against necrotizing enterocolitis and coccidiosis in broilers (Han et al. 2023; Chaudhari et al. 2020). The yeast *Saccharomyces cerevisiae* is abundant in easily absorbable protein, nucleotides, amino acids, B vitamins, enzymes, and other essential nutrients (Sivinski et al. 2022). It exhibits regulatory effects on livestock immunity and antioxidant function while enhancing production performance, and also it has gained extensive utilization in the field of livestock farming (Cattaneo et al. 2023).”

3. The introduction outlines the importance of finding alternatives to antibiotics for improving rabbit health but does not clearly articulate the specific research gap or the novelty of the current study. Including a clearer statement about the gap in the literature that this study aims to address could provide readers with a better understanding of its significance.

- According to your suggestion, we have revised the introduction and provided more details on the uniqueness of this study.

“Although *Saccharomyces cerevisiae*, *Bacillus subtilis*, and *Bacillus licheniformis* have been extensively investigated in livestock breeding, such as rabbits, there is a paucity of research on the combined application of these three probiotics. We hypothesize that supplementation of compound microecological preparations comprising *Bacillus subtilis*, *Saccharomyces cerevisiae*, and *Bacillus licheniformis* during lactation could enhance the performance and intestinal microbiota of lactating female rabbits and their offspring, thereby improving the reproductive efficiency of female rabbits and growth outcomes in offspring. Thus, this study aims to compare the effects of

incorporating varying levels of compound microecological agents into diets and investigate their impact on the performance, immune response, antioxidant properties, and intestinal flora of both mother rabbits and their offspring, in order to provide theoretical evidence for the application of microecological agents in rabbit breeding production.”

Material and methods:

Please address these issues in material and methods section

1. While the study mentions that animals were randomly assigned to groups, it does not provide specific details about the randomization process. Providing information about how randomization was conducted could enhance the transparency and reproducibility of the study

- To ensure the randomness of the experimental female rabbits, we added additional weight data from Table 2. We carefully selected 60 healthy female rabbits with similar weights and mated them simultaneously on the farm to conduct this experiment.

“The study included a total of sixty healthy female rabbits, which were carefully selected based on their similar birth weight and parity (Table 2).”

2. The section describing sample collection for both young rabbits and female rabbits lacks detailed information about standardized procedures followed during sample collection. Providing specific details about standardized protocols for sample collection could ensure consistency and reliability of the data.

- Thank you for your valuable suggestions, which have greatly enhanced the rigor and perfection of our article. We have further enriched the sampling process by conducting a thorough literature review and incorporating multiple revisions.

“For young rabbits, after an overnight fasting period at 14 and 35 days of age, four young rabbits were randomly selected from each group for sample collection. Specifically, four litters of rabbits were randomly chosen in each group, and individuals with weights close to the average weight of their respective litter were selected. At the end of the experiment, blood was collected from the heart of the young rabbits, and immediately injected into the heparin sodium vacuum blood collection tube after blood collection with a syringe, and mixed upside down for 7-8 times (Cao 2020).”

“Most of the healthy rabbits exhibit fecal pellets in their intestinal tract. The specific procedure for collecting feces from female rabbits is as follows: prior to the morning feeding at 8:00, one hand firmly grasps the rabbit's ears and neck fur, while the palm of the other hand supports its buttocks, causing a slight elevation of the rabbit's abdomen. By gently separating the hind limbs, the anus is exposed. Two fingers are then applied with gentle pressure approximately 5 cm from the anus towards it, resulting in extrusion of 2-3 fresh rabbit fecal pellets. And then the fresh fecal samples were loaded into 5ml frozen tubes, rapidly frozen in liquid nitrogen, and subsequently stored at -80°C for further analysis of gut microbiota.”

3. Although the statistical analysis is mentioned, the section lacks specific details about

the types of statistical tests used and how the data were analyzed. Providing more information about the statistical methods employed could help readers better understand the robustness of the study's findings.

- According to your guidance and suggestions, we have reedited the data analysis section.

“The data analysis of this study employed a diverse range of statistical methods to ensure the comprehensiveness and accuracy, thereby enhancing the professional and academic quality of the research. The experimental data were initially processed in Microsoft Excel 2019 software, and then the statistical significance was experimentally tested using one-way ANOVA in IBM SPSS Statistics 27 software (SAS Inc., Chicago, IL), differences between groups were compared using the least significant difference (LSD) method, with ($P < 0.05$) considered significant and ($P < 0.01$) highly significant (Park et al. 2009). The results are presented as the mean \pm standard error (SEM), and multiple comparisons were corrected using Duncan's correction (Su et al. 2022).

It is worth noting that the sample size in this study was limited, which may have implications for detecting subtle effects. Furthermore, the experimental design focused on a specific probiotic formulation and dose, potentially limiting its generalizability to all production environments or rabbit populations. Future studies should consider larger sample sizes and more comprehensive experimental designs to validate these preliminary findings and explore the potential impact of compound probiotics on other animal populations.”

Results

1. The results do not include comparisons with baseline values or references to normal ranges for the measured parameters in rabbits. This makes it challenging to interpret the significance of the observed changes without understanding the starting point.

- Thank you for your inquiry about this part. We are delighted to provide you with an answer. The main objective of this study is to investigate the effects of a compound microecological preparation on the performance, immune and antioxidant indicators, as well as intestinal flora structure of lactating rabbits and rabbits in order to explore its potential benefits on various growth and development parameters in experimental rabbits. For this experiment, we selected 60 healthy and normal rabbits who were randomly divided into four groups. At the beginning of the experiment, all rabbits in each group were in good health; throughout the experiment, we meticulously recorded the elimination rate and health status of each group; after completion, we used data from the control group (group without any treatment) as a reference range for rabbit measurement parameters to determine whether supplementation with compound microecological preparation had a promoting effect on both lactating rabbits and their offsprings.

2. While statistical tests are mentioned for some comparisons, it's unclear whether adjustments were made for multiple comparisons. Without proper correction, there's a risk of false positive results, especially when analyzing multiple biochemical markers and immune indexes.

- The analysis method indicated in the data analysis section of this paper was strictly followed for all the test data analysis conducted in this study: “The data analysis of this study employed a diverse range of statistical methods to ensure the comprehensiveness and accuracy, thereby enhancing the professional and academic quality of the research. The experimental data were

initially processed in Microsoft Excel 2019 software, and then the statistical significance was experimentally tested using one-way ANOVA in IBM SPSS Statistics 27 software (SAS Inc., Chicago, IL), differences between groups were compared using the least significant difference (LSD) method, with ($P < 0.05$) considered significant and ($P < 0.01$) highly significant (Park et al. 2009). The results are presented as the mean \pm standard error (SEM), and multiple comparisons were corrected using Duncan's correction (Su et al. 2022).”

3. The sample size and number of replicates are mentioned ($n=4$), which may be considered low for drawing robust conclusions, particularly in complex biological systems like the gut microbiota. So authors could add a sentence for this potential shortcoming that "Larger sample sizes would enhance the reliability of the findings".

- Thank you so much for your suggestion! We've taken it on board and added a sentence to the article explaining the limitations of our sample size in this experiment.

“SEM = standard error of the mean ($n = 4$, number of replicates; larger sample sizes would enhance the reliability of the findings)”

4. The results do not mention whether potential confounding factors, such as age, sex, or health status of the rabbits, were controlled for in the analysis. Failure to account for these variables could introduce bias into the results.

- Thank you for your inquiries regarding this section. We are delighted to provide answers for you. At the beginning of the study, we carefully selected lactating female rabbits with similar age and weight, aiming to minimize the impact of age and physiological status on the study results. Specifically, at 24 days of gestation, we meticulously chose 60 healthy female rabbits with similar gestation periods and weights to ensure consistency among the study groups. Moreover, our research solely focused on lactating female rabbits and their offspring. By design, we exclusively assessed the effects of compound microbial preparation on lactation performance and offspring growth, thus effectively controlling for gender variables through this selection criterion. Throughout the study duration, we closely monitored both the health status of lactating female rabbits and their offspring. The inclusion criteria required all participating rabbits to be in good health so as to minimize any potential confounding effects caused by pre-existing health conditions. The content mentioned above has been thoroughly explained in the experimental design section of this article.

Discussion

1. The discussion predominantly focuses on summarizing the results and existing literature without providing a critical analysis or interpretation of the findings. A more in-depth discussion that critically evaluates the implications of the results in the context of the research field could enhance the manuscript's quality.

- Thank you so much for bringing to our attention the areas where the discussion section of this paper could be improved. With your guidance and our understanding of this study, we have restructured and rewritten the discussion section.

2. While the discussion briefly mentions previous studies that support the findings, there

is a lack of detailed comparison with existing literature. Providing a more comprehensive comparison with previous research, including both similarities and differences, could strengthen the discussion and highlight the novelty of the current study.

- We have reedited the discussion section of this article.

3. Some points in the discussion are repeated, leading to redundancy and lack of conciseness. Streamlining the discussion and avoiding repetition would improve clarity and readability.

- Thank you so much for your suggestion. We have revised the discussion section of this article and eliminated any unnecessary parts.

Conclusion

The conclusion fails to provide a clear recommendation or actionable insight based on the study's findings. While it mentions the potential benefits of compound microecological preparations, it does not offer specific guidance or practical implications for future research or application in rabbit husbandry practices.

- Currently, there is limited research on compound microecological preparations for rabbits, as they are primarily used in dairy cows, pigs, sheep, and other livestock. However, this study demonstrates that incorporating compound microecological preparations can enhance the lactation performance and overall growth and development of lactating rabbits, which contributes significantly to the healthy advancement of the rabbit industry. In practical breeding scenarios, it is recommended to mix the compound microecological preparations from this study into the feed based on the actual intake of lactating rabbits at each farm.

“Our findings highlight the potential of compound microecological preparations as an effective strategy for enhancing lactation performance, immune function, and antioxidant capacity in rabbits. The supplementation of probiotics through RBM offers a promising approach to optimize the growth and health outcomes of newborn rabbits. Further research is required to elucidate the underlying mechanisms and explore the long-term effects of compound microecological preparations on rabbit health and production performance. In all, this study suggests that the inclusion of compound microecological preparations at a dosage of 6g/d in the diets of lactating rabbits can effectively enhance both performance and health status of lactating rabbits and their offspring. Additionally, the practical implications of our study extend to the development of more targeted probiotic formulations, enhancing animal welfare and productivity in sustainable farming practices. Future research should focus on longitudinal studies to ascertain the long-term impacts of such interventions, including potential generational effects.”

Please answer to these question positively

1. What were the effects of compound microecological preparations on the plasma biochemical markers of rabbits?

- Thanks for your question. In this study, no significant differences were observed in plasma biochemical indexes between the experimental and control groups of lactating female and young rabbits, and we speculate that the supplementation of compound probiotics did not lead to an

increase in the body burden of experimental rabbits and had no discernible impact on their overall health.

2. Were there any significant differences observed in the biochemical indicators between the experimental and control groups of female rabbits?

- Thanks for your question. In this experiment, our results showed that the addition of compound microecological preparations had no effect on the plasma biochemical indexes of lactating female rabbits, and also our proposed explanation for this result is that compound probiotics did not lead to an increase in the body burden of experimental female rabbits and had no discernible impact on their overall health.

3. Can you provide information on the immune and antioxidant indexes of rabbits after the administration of compound microecological preparations?

- Thanks for your question. The immune indicators tested in this study included immunoglobulin G (IgG), immunoglobulin A (IgA), immunoglobulin M (IgM), secretory immunoglobulin A (sIgA), interleukin-2 (IL-2), interleukin-6 (IL-6), interleukin-8 (IL-8), tumor necrosis factor- α (TNF- α) and antioxidant indicators include catalase (CAT), glutathione peroxidase (GSH-Px), superoxide dismutase (SOD) and malondialdehyde (MDA).

The results show that: supplementation with a combination of microecological agents can significantly enhance the immune and antioxidant properties of both female rabbits and their offspring as a whole, but only certain plasma indexes in young rabbits are significantly affected by indirect intake of breast milk. We speculate that this could be due to inadequate intake of complex microecological agents in young rabbits.

“The impact of incorporating compound microecological preparations into the diet of female rabbits on immune indexes is illustrated in Table 4. Compared to the control group, the plasma concentrations of IgG in groups A, B, and C exhibited significant increases of 27.48%, 41.09%, and 29.04% respectively; the concentrations of IgA showed significant increases of 13.62%, 21.74%, and 22.94% respectively; and the concentrations of IgM demonstrated significant increases of 21.46%, 21.33%, and 15.21% respectively ($P < 0.05$). Compared to the control group, the concentrations of sIgA in the plasma of groups A, B, and C exhibited significant increases of 20.84%, 46.89%, and 26.67% ($P = 0.028$), respectively. In terms of cytokines, the addition of compound microecological preparations to the rabbit diet significantly increased the concentrations of anti-inflammatory cytokines IL-2 and IL-8 ($P < 0.05$), while decreasing the levels of pro-inflammatory cytokines IL-6 and TNF- α . Among all groups, group B exhibited the most pronounced effect in elevating anti-inflammatory cytokine concentrations and reducing pro-inflammatory cytokine contents in plasma. Compared to the control group, group B showed a significant increase of 43.67% and 40.28% in plasma concentrations of anti-inflammatory cytokines IL-2 and IL-8, respectively, along with a significant decrease of 8.28% and 12.23% in contents of pro-inflammatory cytokines IL-6 and TNF- α . In terms of antioxidant indexes, the supplementation of compound microecological preparation did not have a significant impact on the content of SOD in plasma ($P > 0.05$), while it significantly affected the concentrations of other antioxidant indexes, including GSH-PX, CAT, and MDA ($P < 0.05$). The concentrations of

GSH-PX and CAT in the plasma of female rabbits in group B were significantly increased by 36.97% and 28.12%, respectively, and the concentration of MDA was significantly decreased by 36.86%. In terms of improving the immune performance and antioxidant performance of female rabbits, the group B achieved the best effect overall, and groups A and B showed advantages only in some indexes of IgA and IgM.

For young rabbits, compared to the control group, the experimental groups A and C exhibited a significant increase in plasma IgA concentration by 7.17% and 1.29% respectively ($P=0.024$), and the experimental group A showed a significant elevation of 9.97% in plasma sIgA concentration ($P=0.023$). In terms of cytokines, the indirect administration of compound microecological preparations through rabbit milk as a medium significantly increased the plasma concentration of anti-inflammatory cytokine IL-8 in young rabbits ($P<0.001$), and this increase was observed to be 47.25% and 13.16% in groups B and C, respectively. Furthermore, it led to a significant reduction in the plasma content of pro-inflammatory cytokine IL-6 by 22.43%, 6.91%, and 10.58% in groups A, B, and C, respectively ($P=0.003$). Additionally, there was an inclination towards reducing the concentration of pro-inflammatory cytokine TNF- α in plasma ($P=0.059$). In terms of antioxidant indexes, the inter-group effects of compound microecological preparations indirectly administered to rabbits through breast milk did not show significant changes in the levels of SOD, MDA, and GSH-PX in plasma ($P>0.05$). However, there were statistically significant effects on the concentration of CAT in plasma, compared to the control group, the groups A, B, and C exhibited a significant increase in CAT concentration by 28.15%, 13.30%, and 8.26% respectively ($P=0.001$). The findings suggest that the indirect administration of compound microecological preparations through rabbit milk exerts a significant impact on certain immune and antioxidant markers in young rabbits, while also effectively enhancing their immune and antioxidant capacity to a certain extent (Table 4).”

4. What were the findings regarding the intestinal microbiota of female rabbits after the administration of compound microecological preparations?

- Thanks for your question. The results of this study demonstrated that supplementing compound microecological preparations significantly enhanced the structure of intestinal flora in female rabbits at the phylum, family, and genus levels.

“Both Firmicutes and Bacteroidetes play a crucial role in the rabbit intestine, with Firmicutes demonstrating greater stability compared to Bacteroidetes. Consequently, an increased ratio of Firmicutes to Bacteroidetes contributes to enhanced protection against harmful bacteria, thereby promoting intestinal health. In female rabbits, the abundance of Firmicutes was found to be higher in group B than in the control group, aligning with previous research findings. The findings of this study further demonstrate that the administration of compound microecological preparations to lactating female rabbits can significantly enhance the abundance of Rikenellaceae in their gut microbiota. Previous studies have indicated that the Rikenaceae_RC9_gut_group may play a crucial role in the breakdown of crude fiber, while the Rikenella_RC9_gut_group genus is essential for feed absorption and utilization in Tan sheep. Therefore, we hypothesize that Rikenella_RC9_gut_group enhances the tricarboxylic acid cycle activity in female rabbits by accelerating glucose production through dietary cellulose degradation. This mechanism ensures

sufficient energy generation to meet the body's requirements and reduces reliance on external energy sources.”

Re: Spectrum00067-24R1 (Dietary supplementation with compound microecological preparations: Effects on the production performance and gut microbiota of lactating female rabbits and their litters)

Dear Prof. Zhanjun Ren:

Your manuscript has been accepted, and I am forwarding it to the ASM production staff for publication. Your paper will first be checked to make sure all elements meet the technical requirements. ASM staff will contact you if anything needs to be revised before copyediting and production can begin. Otherwise, you will be notified when your proofs are ready to be viewed.

Sincerely,
Francisco Uzal
Editor
Microbiology Spectrum